# NORADRENERGIC-INSPIRED GAIN MODULATION ATTENUATES THE STABILITY GAP IN JOINT TRAINING

## ABSTRACT

Recent work in continual learning has highlighted the *stability gap* – a temporary performance drop on previously learned tasks when new ones are introduced. This phenomenon reflects a mismatch between rapid adaptation and strong retention at task boundaries, underscoring the need for optimization mechanisms that balance plasticity and stability over abrupt distribution changes. While optimizers such as momentum-SGD and Adam introduce implicit multi-timescale behavior, they still exhibit pronounced stability gaps. Importantly, these gaps persist even under ideal joint training, making it crucial to study them in this setting to isolate their causes from other sources of forgetting. Motivated by how noradrenergic bursts transiently increase neuronal gain under uncertainty, we introduce a dynamic gain scaling mechanism as a two-timescale optimization technique that balances adaptation and retention by modulating effective learning rates and flattening the local landscape through an effective reparameterization. Across domain- and class-incremental MNIST, CIFAR, and mini-ImageNet benchmarks under task-agnostic joint training, dynamic gain scaling effectively attenuates stability gaps while maintaining competitive accuracy, improving robustness at task transitions.

## 1 INTRODUCTION

Continual learning (CL) in artificial neural networks aspires to emulate this lifelong adaptability – networks acquire incoming data continuously, without *catastrophically* overwriting the knowledge of past examples (Van De Ven et al., 2022; Kudithipudi et al., 2022; Mei et al., 2025). Traditionally, most work in this field has concentrated on mitigating this phenomenon by approximating the ideal joint loss over all previous tasks encountered (Van De Ven et al., 2022; De Lange et al., 2022). However, recent research has pointed out the presence of *stability gaps* (De Lange et al., 2022; Hess et al., 2023) – a transient forgetting of past task knowledge at every task switch (Figure 1). Crucially, this gap remains even under ideal joint training, indicating that it originates from the *dynamics of optimization* rather than from representational drift or limited memory capacity (Hess et al., 2023; Kamath et al., 2024).

Standard optimizers such as momentum-SGD (MSGD) (Qian, 1999) and Adam (Kingma & Ba, 2014) introduce implicit multi-timescale regularization through momentum or adaptive learning rates, yet they still exhibit stability gaps. Recent analyses show that momentum terms can deviate the update from the true gradient direction (Kamath et al., 2024), causing overshooting when the loss landscape shifts abruptly at task boundaries. This makes the stability gap a dynamics-induced issue that remains unresolved, and one that is particularly detrimental in *online* CL, where robustness throughout training is essential.

However, in biology, neural circuits demonstrate lifelong adaptability, continually integrating new information with minimal disruption to established memories (Kudithipudi et al., 2022). This finely tuned balance between plasticity and stability arises from intrinsic neuromodulatory dynamics in cortical networks (Mei et al., 2025; Shine, 2019). A key mechanism supporting this flexibility is gain neuromodulation – transient changes in neuronal responsiveness that adjust how strongly inputs influence neural activity without altering selectivity (Ferguson & Cardin, 2020; Jordan, 2024). Although gain modulation can arise through multiple biological pathways, a well-characterized source is noradrenergic activity that increases neural gain in response to unexpected uncertainty during sen-

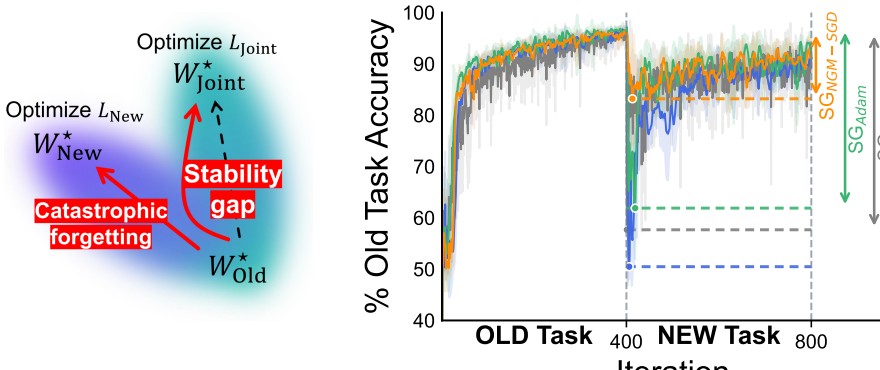

Figure 1: **Schematic of the stability gap.** *Left panel.* Conceptual illustration clarifying how the stability gap differs from catastrophic forgetting: optimizing a new task can induce a temporary drop in old-task performance even without erasing previously learned solutions. *Right panel.* Old-task accuracy across two sequential tasks on Split CIFAR-10, showing optimizer-specific performance drops at the task switch and the resulting stability gaps.

sory evidence integration, enabling rapid updates to evolving perceptual estimates while preserving stable representations (Aston-Jones & Cohen, 2005; Shine, 2019; Munn et al., 2021).

We hypothesize that these transient gain boosts induce fast and slow timescales of weight adaptation, enabling rapid adjustments to new information while preserving stable representations, analogous to the fast–slow weight optimization paradigm – where fast weights momentarily improve performance on earlier tasks during new learning without disrupting the consolidation occurring in slow weights (Hinton & Plaut, 1987). In addition, gain modulation implicitly induces a parameter reparameterization that flattens the loss (energy) landscape, providing further stability by reducing sensitivity to abrupt changes (Wainstein et al., 2025).

Since efforts to understand the stability gap lead to the realization that we should focus not only on *what* to optimize, but also on *how* to optimize (i.e. the optimization technique (Hess et al., 2023), see Figure 1), we introduce an uncertainty-modulated gain mechanism to act as an approximate two-timescale gradient descent adjustment to mitigate stability gaps in *task-agnostic online* CL scenarios. We benchmark against Adam (Kingma & Ba, 2014) and MSGD (Qian, 1999), as both are widely used multi-timescale optimizers, while also including vanilla SGD (Goodfellow et al., 2016) as a non-adaptive single-timescale baseline. Thus, the main contributions of our work are:

- We introduce uncertainty-guided gain dynamics and analytically show that transient gain boosts in gradient descent updates induce emergent fast and slow weight-adaptation timescales, mirroring multi-timescale optimization methods.

- We demonstrate that scaling neuronal gain yields an effective reparameterization of the weight space that locally flattens the loss landscape, reducing interference during task transitions.

- Building on these insights, we introduce a two-timescale optimizer for *online* CL training which effectively attenuates the stability gap. We empirically validate our approach on both domain-incremental and class-incremental MNIST (Split MINST, Rotated MNIST), CIFAR (Split CIFAR-10, Domain CIFAR-100), and ImageNet (split mini-ImageNet) benchmarks under task-agnostic joint training, demonstrating attenuation in the stability gap compared to the MSGD, Adam, and vanilla SGD optimizers.

Note that as our contribution concerns *how* the model is optimized, and despite our tests rely on an idealized joint training setting, NGM-SGD is fully compatible with existing continual learning interventions that address catastrophic forgetting such as replay methods and regularization techniques.

## 2 RELATED WORK

**Stability gaps in CL.**   It was not until recently that a study pointed out the conventional evaluation paradigm for AI systems is fundamentally inadequate (De Lange et al., 2022; Hess et al., 2023). Consider a sequence of tasks $\{T_k\}_{k=1}^K$ with corresponding data distributions $\{D_k\}_{k=1}^K$ and let $\mathcal{F}_\theta$ denote a model parameterized by $\theta$. Under the traditional task-based evaluation, $\mathcal{F}_\theta$ is trained sequentially, updating its parameters $\theta_k$ upon completing task $T_k$, and its performance is measured only immediately after training on $T_k$ has finished (De Lange et al., 2022). Notably, this task-based evaluation is blind to how a model's accuracy evolves during training. As an alternative, the continual evaluation strategy measures the performance of $\mathcal{F}_\theta$ every $\rho$ training iterations while learning task $T_k$ (De Lange et al., 2022). Interestingly, continual evaluation revealed stability gaps (De Lange et al., 2022; Caccia et al., 2021) even in large language models (Guo et al., 2024). These brief accuracy drops undermine the goals of CL by introducing temporary forgetting between tasks and limiting the system's overall learning capacity. Specifically, this might be especially problematic in continual safety-critical applications, where even momentary failures can have severe consequences (De Lange et al., 2022; Hess et al., 2023). One may naively attribute the stability gap to imperfect approximations of the joint objective by CL methods. Yet this hypothesis does not hold up as the phenomenon persists even under perfect joint training, where $L_{\text{joint}} = L_{\text{new}} + L_{\text{old}}$ (Hess et al., 2023; Kamath et al., 2024). Hence, this finding implies that the transient performance collapse is not merely an artifact of memory or regularization approximations, but rather emerges from the fundamental dynamics of sequential optimization, underscoring the importance of studying it in the idealized joint-loss setting (Kamath et al., 2024).

Beyond evaluation paradigms and identification of the stability gap, a growing body of work has sought to pinpoint its causes and propose remedies. Hess et al. (2023) argue that there exists a non-increasing-loss path between $\theta_{\text{old}}$ and $\theta_{\text{joint}}$, suggesting that a continual learner only needs to follow this valley. However, enforcing such trajectories via gradient-projection methods has shown limited practical success. In a homogeneous CIFAR setting, Kamath et al. (2024) further demonstrate that although a linear low-loss path does exist, MSGD consistently deviates from it at each task transition. Although successful approaches to mitigate the stability gap exist, many rely on added architectural modules or task-specific details, making them impractical for lightweight, task-agnostic online use. For example, Łapacz et al. (2024) enlarge the classifier with each new task – leading to growing model complexity – and demonstrate that much of the transient forgetting originates in the final linear layer. Harun & Kanan (2023) and Guo et al. (2024) rely on pretrained backbones and low-rank updates, and Soutif-Cormerais et al. (2023) maintain a smoothed average of the weights across task transitions, improving stability at computational cost. These limitations highlight the need for lightweight, optimiser-level interventions compatible with online task-agnostic training. Our method takes precisely this route, introducing a bio-inspired gain-modulated update rule to mitigate the stability gap through the update dynamics.

**Neuronal gain modulation.**   Neuronal gain characterizes the input sensitivity of a neuron. It is formally defined as the slope of its input-output transfer function, reflecting the rate at which output firing rates change in response to variations in input current (Ferguson & Cardin, 2020; Jordan, 2024; Aston-Jones & Cohen, 2005). This property has been extensively investigated in biophysical models of cortical circuits, highlighting the role of gain modulation in adaptive computation (Munn et al., 2023; Bos et al., 2025), and fine-tuning motor responses (Stroud et al., 2018). Furthermore, theoretical neuroscience works have debated the causal relationship between gain modulation and synaptic plasticity, suggesting a complex interplay (Swinehart & Abbott, 2005). However, despite its biological relevance, gain modulation has received limited attention in artificial learning systems. A recent study has used gain modulation for attentional gating in hierarchical models but only as a fine-tuning step *after* weight learning, ignoring its interaction with weight updates (Caroline Haimerl et al., 2022). Another study used it as an isolated mechanism for adaptive whitening (Duong et al., 2023a) and extended it to multi-timescale whitening by pairing fast gain changes with slower synaptic plasticity (Duong et al., 2023b). However, their gain modulation was limited to interneurons in a single-layer architecture (Duong et al., 2023a;b). Nevertheless, a key theoretical insight is that gain neuromodulation reshapes the energy landscape: noradrenergic input flattens it, thus enabling flexible switching, while cholinergic input deepens it to stabilize memories (Munn et al., 2021; 2023; Shine, 2019). Recent recurrent neural network work supports this view, showing that uncertainty-driven gain modulation facilitates perceptual switches by propelling the system

through high-velocity trajectories (Wainstein et al., 2025). However, in that model, gains were fixed during training and modulated only afterward, limiting interaction with weight updates. We extend this approach by including gain dynamics during learning and applying it to reduce stability gaps in joint CL.

## 3 GAIN BOOSTS AS A MULTI-TIMESCALE OPTIMIZER

Introducing processing timescales into artificial neural networks provides a principled strategy to reconcile stable, long-term memory consolidation with rapid, context-driven adaptation. The fast-slow weight paradigm exemplifies this principle by maintaining distinct "slow" and "fast" weight components, i.e. $W_{\text{eff}} = w_{\text{slow}} + w_{\text{fast}}$ (Hinton & Plaut, 1987). This enables networks to swiftly adapt to distribution shifts via transient updates in the fast weights, which rapidly decay toward zero, while simultaneously preserving consolidated knowledge within slow weights (Jones, 2022). Crucially, this mechanism minimizes interference between newly acquired information and existing representations (Hinton & Plaut, 1987), making it particularly suitable as a foundation for lifelong learning in artificial systems.

In biological neural circuits, analogous multi-timescale processing occurs through neuromodulatory signals that adjust neuronal gain, thereby altering neuronal responsiveness to incoming stimuli (Ferguson & Cardin, 2020; Shine, 2019; Aston-Jones & Cohen, 2005). For instance, during perceptual updating, the noradrenergic system originating from the locus coeruleus transiently boosts neuronal gain to facilitate sensory integration of new incoming stimuli (Wainstein et al., 2025; Jordan, 2024; Aston-Jones & Cohen, 2005). We propose that this neuronal gain modulation naturally gives rise to a fast-slow learning structure by shaping the effective synaptic weights. Consider a model $y(t) = h(x(t); g(t), w(t))$ and a loss function $L(y(t), T(t))$, where $x(t)$ represents the input at time $t$, $\{g(t), w(t)\}$ denotes the parameter estimate, $y(t)$ is the output produced by the model, and $T(t)$ is the target. In artificial neural networks, we can interpret $w_{ij}(t)$ as the synaptic weights connecting neuron $j$ with neuron $i$ and $g_i(t)$ as the neuronal gain; hence, one can consider an effective weight

$$W_{ij}(t) = g_i(t)w_{ij}(t). \tag{1}$$

Assuming neuronal gain follows noradrenergic activity, we distinguish a phasic mode of rapid bursts driven by prediction errors that transiently boost gain, and a tonic baseline mode $g_0$. This leads to a decomposition of the effective weight as contributions from these components, i.e.

$$W_{ij}(t) = g_i(t)w_{ij}(t) = g_0 w(t) + [g_i(t) - g_0] w_{ij}(t). \tag{2}$$

This decomposition highlights how neuronal gain modulation inherently realizes a two-timescale optimization scheme. The slow component, $w_{ij}^{\text{slow}}(t) = g_{0,i}w_{ij}(t)$, represents the stable synaptic changes associated with tonic neuromodulation. Conversely, the fast component, $w_{ij}^{\text{fast}}(t) = [g_i(t) - g_0] w_{ij}(t)$ dynamically emerges in response to phasic neuromodulatory signals, effectively isolating rapid contextual adaptations from stable memory traces. Simply, transient gain modulation "virtually" decouples the effective synaptic weight updates into slow and fast components, offering an adaptive mechanism to flexibly balance plasticity and stability (Figure 2A). To further illustrate this concept, following Jones (2022), we consider a simplified linear model defined by $y(t) = W(t)x$ with a constant input $x \equiv 1$ to predict a target signal $T(t)$ under the mean square error loss $L = \frac{1}{2}[T(t) - y(t)]^2$. Parameter updates follow standard stochastic gradient descent, while gain dynamics evolve via exponential decay modulated by transient increases in response to contextual surprises (Wainstein et al., 2025) – see Appendix F for a further discussion. Formally:

$$w(t+1) = w(t) - \alpha\partial_w L, \tag{3}$$
$$g(t+1) = \gamma g(t) + (1-\gamma)g_0 + \eta H(y) \tag{4}$$

where $\alpha$ is the learning rate, $\gamma < 1$ controls how quickly the gain decays back to its tonic baseline ($g_0 = 1$ for simplicity), and $\eta$ determines the strength of the phasic bump, driven by the "surprise" signal $H(y)$, here approximated by the absolute value of the loss gradient $|\partial_g L|$, reflecting its link to error prediction (Jordan, 2024). Given that the gradient with respect to synaptic weights can be expressed as $\partial L/\partial w = g(t)(\partial L/\partial W)$, neuronal gain acts explicitly as a plasticity modulator, dynamically scaling the magnitude of learning updates. During high uncertainty or unexpected errors, phasic increases in neuronal gain amplify synaptic weights during backpropagation, allowing

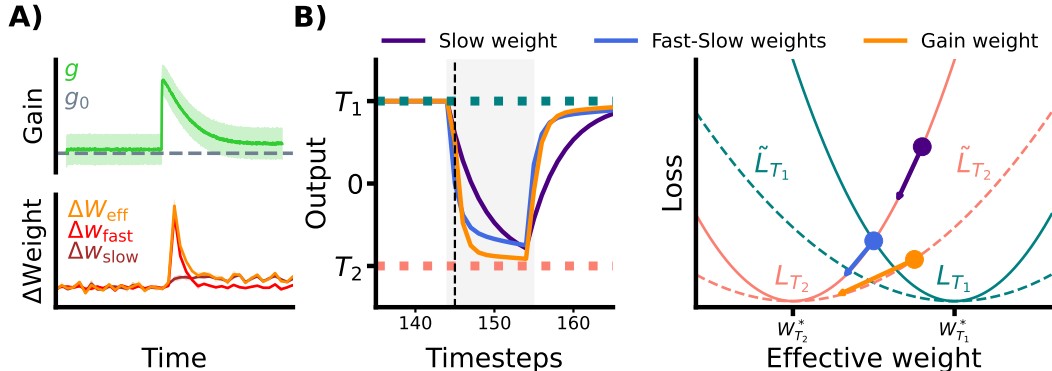

Figure 2: **Simple proof of principle of gain boost approximating a two-timescale optimizer.**
**A)** Schematic of gain-induced effective weight decomposition under noradrenergic neuromodulation. Phasic noradrenergic signals transiently increase neuronal gain, thereby splitting the effective weight into a fast–slow scheme with $w_{\text{fast}} = [g(t) - g_0]\, w(t)$ and $w_{\text{slow}} = g_0\, w(t)$. **B)** *Left panel.* Simplified model $y(t) = W_{\text{eff}}(t)\, x$ with constant input $x$ predicting a target $T(t)$ under mean square error loss $L = \frac{1}{2}[T(t) - y(t)]^2$. Comparison of gain-modulated (orange) and fast–slow (blue) and slow (purple) weight methods. The unshaded region corresponds to optimization toward target $T_1$, and the shaded region to optimization toward target $T_2$. The dashed vertical line marks the time of peak neuronal gain. *Right panel.* Loss landscape flattening induced by gain neuromodulation at the highest neuronal gain. In the effective weight space, gain boosts reparametrize the loss, effectively reducing its curvature, $\lambda \to \lambda/g^2$. Dots illustrate each method's state at the peak gain time, and arrows indicate the gradient step.

rapid adaptation to novel contexts. Moreover, these gain boosts induce a reparameterization of the loss (Dinh et al., 2017), under which the curvature of the energy landscape in the effective weight space is reduced as $\lambda \to \lambda/g^2$ (see Appendix B for a theoretical demonstration). Once confidence is restored, gain returns to baseline, promoting stability. Thus, gain modulation not only amplifies plasticity but also transiently flattens the loss surface, facilitating smoother transitions between distinct attractors (Figure 2B, left panel).

## 4 JOINT TRAINING EXPERIMENTS

Our hypothesis is rooted in gain modulation's ability to both segregate synaptic plasticity into fast and slow adaptation channels and flatten the network's energy landscape. This dual action should narrow stability gaps between successive tasks and lower test-set loss at contextual changes, together minimizing interference. Thus, to rigorously understand the impact of our approach in attenuating the stability gap, we isolate the stability gap from other sources of forgetting by using a joint-training experimental setting. We consider a scenario in which a learner $F_\theta$ is trained on a sequence of $K$ contexts $\mathcal{C} = \{c_k\}_{k=1}^K$. Each context $c_k$ is a disjoint dataset $\{(\mathbf{x}_j, \tilde{y}_j)\}_{j=1}^N \sim \mathcal{D}_k$. Contexts are presented one after another in an online fashion, with each context trained for a fixed number of iterations $I_\mathcal{C}$ [1]. Since this is a joint-training setup, the model retains access to all data from previous contexts throughout learning.

**Noradrenergic uncertainty.** Biological and theoretical evidence converge on the necessity of explicitly representing a learner's uncertainties when interacting with stochastic, non-stationary environments to achieve optimal inference (Yu & Dayan, 2005; Dayan & Yu, 2006; Jordan, 2024). Noradrenaline has been particularly linked to "unexpected uncertainty", where its phasic signaling is triggered by sudden increases in environmental variability due to contextual changes (Yu & Dayan, 2005; Dayan & Yu, 2006; Mei et al., 2025; Jordan, 2024). This response injects additional variability into neural and behavioral processes, facilitating rapid adaptation to shifting contexts (Wainstein

---

[1]Due to random sampling, some training images may be encountered more than once, although data augmentation (crops and flips) ensures that repeated samples may differ across occurrences.

et al., 2025; Munn et al., 2021; Shine, 2019). Evidence in this regard from Wainstein et al. (2025) suggests that this form of uncertainty can be quantified as the entropy of the network's readout:

$$H(y) = \sum_i \pi_i(y) \log \left( \pi_i(y) \right) \tag{5}$$

where $\pi(y)$ denotes the softmax probability distribution of the network's output. We incorporate this into Equation 4 to dynamically modulate neuronal gain, which acts as a forcing function that transiently boosts gain in response to novel or ambiguous stimuli (Jordan, 2024; Aston-Jones & Cohen, 2005; Shine, 2019). To formalize this process, we present Algorithm 1 and benchmark it against MSGD (Qian, 1999), Adam (Kingma & Ba, 2014), and vanilla SGD (Goodfellow et al., 2016). MSGD and Adam implicitly embody multi-timescale dynamics – MSGD can be interpreted as a fast-slow separation where an exponentially decaying gradient trace smooths high-frequency noise (Jones, 2022), while Adam adapts step sizes based on local gradient statistics. These properties make them appropriate baselines for our two-timescale perspective. Vanilla SGD is included as a non-adaptive single-timescale baseline.

---

**Algorithm 1** Noradrenergic Gain-Modulated SGD (NGM-SGD)

---

**Require:** Dataset sequence $\mathcal{C} = \{c_k\}_{k=1}^K$, where each $c_k = \{(\mathbf{x}_j, \tilde{y}_j)\}_{j=1}^N \sim \mathcal{D}_k$; context iterations $I_\mathcal{C}$; batch size $B$; learning rate (lr) $\alpha > 0$; gain–decay $0 < \gamma < 1$; gain-baseline $g_0 \geq 1$; entropy scale $\eta > 0$

1: **Initialize:** $\mathbf{W} \leftarrow \mathbf{W}_{\text{init}}$; $\quad g \leftarrow g_{\text{init}}$
2: **for** each context $c_i \in \mathcal{C}$ **do**
3:     **for** iteration $i = 1$ **to** $I_\mathcal{C}$ **do**
4:         $(\mathbf{X}, \tilde{\mathbf{Y}}) \sim \mathcal{D}_k^B$               ▷ // mini-batch
5:         $\boldsymbol{\pi} \leftarrow \text{softmax}\left(F_{\mathbf{W}}(\mathbf{X}; g)\right)$     ▷ // forward pass
6:         $\mathcal{L} \leftarrow \frac{1}{B} \sum_{j=1}^B \ell(\pi_j, \tilde{y}_j)$     ▷ // cross-entropy loss
7:         $\mathbf{W} \leftarrow \mathbf{W} - \alpha \nabla_{\mathbf{W}} \mathcal{L}$      ▷ // SGD update
8:         $H \leftarrow -\frac{1}{B} \sum_{j=1}^B \sum_l \pi_{j,l} \log \pi_{j,l}$    ▷ // uncertainty (entropy)
9:         $g \leftarrow \gamma g + (1 - \gamma) g_0 + \eta H$     ▷ // gain update
10:     **end for**
11: **end for**

---

## 4.1 METHODS

**Datasets.** For our class-incremental learning experiments, we used the MNIST (Lecun et al., 1998), CIFAR-10 (Krizhevsky & Hinton, 2009), and mini-ImageNet (Vinyals et al., 2016; Russakovsky et al., 2015) datasets. Split MNIST and Split CIFAR-10 were constructed by dividing each dataset into five sequential tasks, each comprising a disjoint subset of the 10 target classes (Hess et al., 2023). For Split mini-ImageNet, the 100 classes were heterogeneously divided into ten tasks of ten classes each, from which we selected five contexts (Hess et al., 2023). For domain-incremental learning, we considered MNIST (Lecun et al., 1998) and CIFAR-100 (Krizhevsky & Hinton, 2009). We defined Rotated MNIST with three tasks, each using the full MNIST dataset rotated by a fixed angle: $0°, 80°, 160°$. These angles were chosen to maximize stability gaps while avoiding digit ambiguities, such as confusing 6 and 9 when rotated $180°$ (De Lange et al., 2022; Hess et al., 2023). We define Domain CIFAR-100 by organizing tasks according to the coarse semantic categories in CIFAR-100 (Hess et al., 2023). Specifically here, each of the twenty CIFAR-100 super-classes is split across three tasks, with each task containing one class from every super-class (20 classes total) to predict the correct super-class.

**Setup.** We employed continual evaluation with $\rho_{\text{eval}} = 1$ and test on all the testing data of the first task (De Lange et al., 2022). Batch sizes were chosen as 128 for experiments on the MNIST dataset and 256 for experiments on the CIFAR-10, CIFAR-100, and mini-ImageNet datasets. Per experiment, the number of iterations per context ($I_\mathcal{C}$) was selected as 200 for Split MNIST and Split mini-ImageNet, 400 for Rotated MNIST and Split CIFAR-10, and 800 for Domain CIFAR-100. Each experiment was repeated five times with different random seeds. For each metric, we report the mean and the standard error across runs.

**Architectures.** For tasks based on the MNIST dataset, we used a feedforward neural network (FFNN) with two hidden layers of 400 units each (De Lange et al., 2022). All neurons in the hidden and output layers were gain-modulated, where gain represents the slope of the neuron's input-output response curve. We employed the ReLU activation function and removed all biases. For CIFAR-10, CIFAR-100, and mini-ImageNet datasets, the increased visual complexity motivated the use of convolutional layers, so we adopted a slim ResNet-18 backbone (Lopez-Paz & Ranzato, 2017; De Lange et al., 2022). Convolutional networks mirror the brain's hierarchy of feature detectors, but unlike biological neurons, where each has its own input-output curve, CNN feature maps use a shared filter, so uniform scaling cannot match sensitivity at the neuron-level and may disrupt useful feature learning. Thus, we restricted gain modulation to the output layer[2], where it could regulate input sensitivity without disrupting feature selectivity, consistent with biological neurons (Ferguson & Cardin, 2020). This choice is also supported by findings that link the classification layer to the stability gap phenomenon (Łapacz et al., 2024).

**Optimization.** NGM-SGD optimization followed Algorithm 1. To determine its hyperparameters, we conducted a soft sweep over the gain-decay parameter $\gamma \in \{0.85, 0.9, 0.95\}$ and the gain scale $\eta \in [0.1, 0.5]$ (see Appendix D). The choice of $\gamma = 0.9$ ensured that the transient gain decayed slowly enough, allowing the transient flattening effect to persist long enough to stabilize the transition, while remaining within a bounded range to support long-term convergence. Dataset-specific $\eta$ values were set to $\{0.5, 0.4, 0.2, 0.1, 0.1\}$ for Rotated MNIST, Split MNIST, Split CIFAR-10, Domain CIFAR-100, and Split mini-ImageNet, respectively. Restricting $\eta$ to this range kept the instantaneous gain $g(t) = g_0 + \Delta g$ low, which prevented unstable learning dynamics – a constraint that can be grounded in biological settings, as cortical gain is bounded by spiking saturation (Munn et al., 2023) – while allowing to tune it according to the dataset complexity to generate sizable gain transients that reduce the stability gap (see Figure 8A in the Appendix F). For comparison, we consider MSGD (Rumelhart et al., 1986) (momentum of 0.9), vanilla SGD (Goodfellow et al., 2016) and Adam (Kingma & Ba, 2014) (betas of 0.9, 0.99) optimizers finding the best learning rates by sweeping lr $= \{1.0, 0.1, 0.01, 0.001, 0.0001\}$ and find that best ones were 0.1 for vanilla SGD, 0.01 for NGM-SGD and MSGD (0.1 in Rotated MNIST), and 0.001 for Adam (see Appendix D). All other training details were kept identical across optimizers to ensure fair comparison.

## 4.2 RESULTS

In Table 1, we summarize the main experimental results across benchmarks using standard continual evaluation metrics established early in the field (De Lange et al., 2022) – see Appendix A. The final average accuracy (avg-ACC) reflects the model's overall performance by averaging classification accuracy across all tasks at the end of training (Hess et al., 2023). The average minimum accuracy (avg-min-ACC) captures the lowest accuracy reached per task, averaged across tasks (Hess et al., 2023), and serves as a measure of transient forgetting. The worst-case accuracy (WC-ACC) incorporates the minimum accuracy on prior tasks and the accuracy on the current task, offering a measure of the stability-plasticity trade-off (Łapacz et al., 2024; De Lange et al., 2022). Lastly, the stability gap (avg-SG) represents the maximum observed drop in accuracy throughout training (Łapacz et al., 2024; Harun & Kanan, 2023), highlighting how much the model degrades when acquiring new tasks.

**NGM-SGD reduces stability gaps.** Table 1 shows that NGM-SGD consistently reduces the stability gap, although the performance gains are less pronounced than hypothesized in prior work (Hess et al., 2023). In class-incremental tasks (Figure 3, top), the gap is largest at the $T_1 \rightarrow T_2$ transition, where the data distribution doubles in complexity (see Table 2 in the Appendix A); it then diminishes as later tasks alter the loss landscape less drastically. In Split CIFAR-10, forgetting is minimal in tasks 2–4 (animal classes distinct from the initial vehicle task[3]), but reappears when vehicles return in the final task. By contrast, mini-ImageNet classes span diverse domains, masking this domain-specific effect. Domain-incremental tasks (Figure 4, top) always exhibit interference due to overlapping feature spaces. Across both settings, stability gaps are most evident with multi-timescale optimizers such as Adam and MSGD, where momentum-driven updates overshoot at task

---

[2]In the NGM-SGD cases on Slim Resnet18, we trained the backbone with MSGD to maintain the network's stability in learning features with the same learning rate.

[3]Task 1 consists of two vehicle categories from CIFAR-10.

Table 1: **Main quantitative metrics.** For all benchmarks we report (across tasks) the average final accuracy (avg-ACC), average minimum accuracy (avg-min-ACC), average stability gap drop (avg-SG), and the worst-case accuracy (WC-ACC). Highlighted values indicate the best results.

| | | avg-ACC (↑) | avg-min-ACC (↑) | WC-ACC (↑) | avg-SG (↓) |
|---|---|---|---|---|---|
| Split MNIST | NGM-SGD (ours) | $99.015 \pm 0.047$ | $\mathbf{97.570 \pm 0.465}$ | $\mathbf{97.696 \pm 0.373}$ | $\mathbf{0.017 \pm 0.005}$ |
| | MSGD | $98.971 \pm 0.086$ | $72.518 \pm 0.403$ | $77.628 \pm 0.329$ | $0.267 \pm 0.004$ |
| | Adam | $\mathbf{99.088 \pm 0.108}$ | $71.047 \pm 3.753$ | $76.537 \pm 3.003$ | $0.283 \pm 0.038$ |
| | SGD | $98.975 \pm 0.099$ | $95.887 \pm 2.887$ | $96.293 \pm 2.311$ | $0.034 \pm 0.029$ |
| Split CIFAR-10 | NGM-SGD (ours) | $\mathbf{90.630 \pm 1.576}$ | $\mathbf{79.485 \pm 3.875}$ | $\mathbf{80.880 \pm 3.349}$ | $\mathbf{0.134 \pm 0.043}$ |
| | MSGD | $89.434 \pm 2.134$ | $52.828 \pm 3.898$ | $58.348 \pm 3.521$ | $0.425 \pm 0.044$ |
| | Adam | $90.342 \pm 1.579$ | $63.450 \pm 7.839$ | $67.692 \pm 6.335$ | $0.310 \pm 0.088$ |
| | SGD | $89.478 \pm 1.909$ | $68.712 \pm 4.958$ | $71.868 \pm 4.007$ | $0.241 \pm 0.059$ |
| Split mini-ImageNet | NGM-SGD (ours) | $\mathbf{48.312 \pm 2.289}$ | $\mathbf{33.165 \pm 2.435}$ | $\mathbf{36.040 \pm 2.058}$ | $\mathbf{0.300 \pm 0.071}$ |
| | MSGD | $46.824 \pm 2.176$ | $26.775 \pm 2.417$ | $30.416 \pm 2.029$ | $0.409 \pm 0.067$ |
| | Adam | $47.208 \pm 2.044$ | $28.300 \pm 3.371$ | $32.052 \pm 2.768$ | $0.384 \pm 0.077$ |
| | SGD | $42.988 \pm 2.549$ | $28.345 \pm 3.671$ | $30.288 \pm 3.309$ | $0.332 \pm 0.098$ |
| Rotated MNIST | NGM-SGD (ours) | $94.948 \pm 0.121$ | $\mathbf{90.015 \pm 1.451}$ | $\mathbf{91.542 \pm 0.973}$ | $0.054 \pm 0.015$ |
| | MSGD | $95.297 \pm 0.217$ | $84.329 \pm 1.359$ | $87.823 \pm 0.914$ | $0.117 \pm 0.014$ |
| | Adam | $\mathbf{95.777 \pm 0.195}$ | $87.177 \pm 1.060$ | $89.910 \pm 0.721$ | $0.092 \pm 0.011$ |
| | SGD | $94.139 \pm 0.255$ | $89.107 \pm 1.935$ | $90.787 \pm 1.300$ | $\mathbf{0.053 \pm 0.021}$ |
| Domain CIFAR-100 | NGM-SGD (ours) | $68.700 \pm 0.881$ | $\mathbf{53.420 \pm 2.804}$ | $\mathbf{58.290 \pm 1.916}$ | $\mathbf{0.224 \pm 0.042}$ |
| | MSGD | $67.160 \pm 0.994$ | $51.370 \pm 2.234$ | $56.867 \pm 1.556$ | $0.232 \pm 0.037$ |
| | Adam | $\mathbf{69.163 \pm 0.871}$ | $51.615 \pm 2.653$ | $57.323 \pm 1.840$ | $0.256 \pm 0.040$ |
| | SGD | $65.463 \pm 1.204$ | $49.050 \pm 3.177$ | $54.950 \pm 2.229$ | $0.243 \pm 0.052$ |

switches, and abrupt loss-landscape shifts trigger transient forgetting. Vanilla SGD stays closer to the true gradient and generally shows smaller gaps (except in high-noise regimes). NGM-SGD further reduces drops by flattening the effective loss landscape on $T_1$, enabling stable evaluation across tasks while still supporting rapid acquisition of new knowledge (see Appendix C for ablation). Additional analyses in the Appendix G show that resetting the inertia terms of Adam and MSGD at task transitions attenuates the overshooting and reduces stability gaps, although both still underperform the vanilla SGD baseline and our NGM-SGD method. Notably, in CIFAR and mini-ImageNet, gain modulation was applied only to the ResNet-18 head, mitigating stability gaps and supporting the findings of Łapacz et al. (2024).

**Gain-boosts reduce test loss at task boundaries.** Aiming to understand the origin of NGM-SGD's ability to reduce stability gaps, we examined the evolution of the $T_1$ loss across all experiments. As shown in Figures 3 and 4, NGM-SGD consistently lowers the test loss at task transitions compared to Adam and MSGD. We hypothesize that this effect arises from the gain reparameterization applied to the forward pass, which flattens the effective landscape of the test-set loss (Dinh et al., 2017). Notably, the learning trajectory followed by NGM-SGD may not correspond to the non-increasing loss path suggested in recent studies (Hess et al., 2023; Kamath et al., 2024). However, unlike Adam and MSGD – whose momentum terms alter the update direction – NGM-SGD follows the true gradient, as the gain only rescales the step magnitude without deviating from the gradient itself. This, combined with the locally flattened Hessian induced by the gain reparameterization, allows the model to incorporate new information more smoothly and avoid sharp increases in loss at task switches.

**Neuronal gain encodes task complexity.** In NGM-SGD, neuronal gain evolves according to Equations 4 and 5, rising in response to task-related uncertainty and decaying as learning progresses (Wainstein et al., 2025). As shown in the bottom row of Figures 3 and 4, the asymptotic level to which gain decays differs between class-incremental and domain-incremental settings, reflecting distinct patterns of task complexity (see Appendix E). In class-incremental scenarios, gain rises at the beginning of each new task and then decays to a progressively higher baseline. This indicates that the system perceives each new task as increasingly complex, requiring higher sustained sensitivity. In contrast, in domain-incremental tasks, gain also spikes at the onset of each task, but consistently decays to a similar level across tasks. This suggests that most of the complexity is

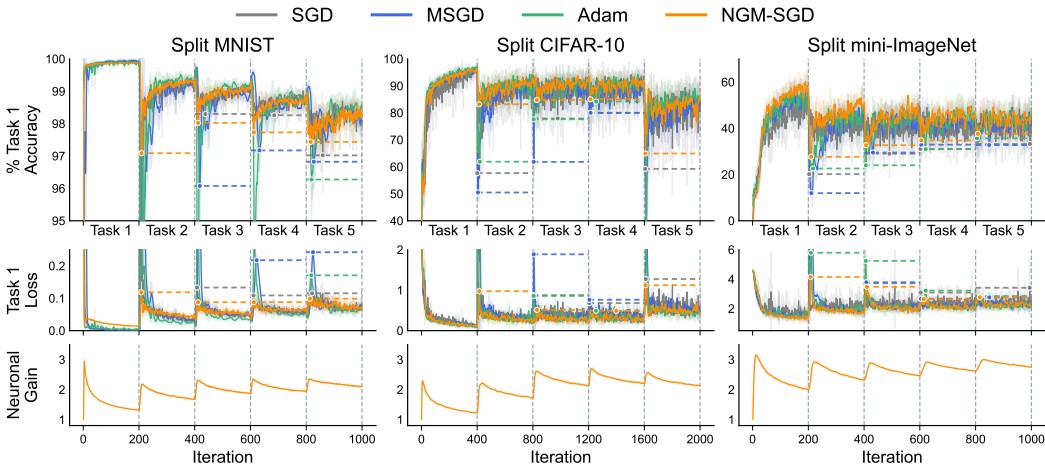

Figure 3: **Stability gaps under class-incremental learning.** The left column shows results on Split MNIST, the center one on Split CIFAR-10, and the right column on Split mini-ImageNet. The top panels display the test accuracy on the first task as the model is incrementally trained on all benchmark tasks. The middle panels show the corresponding test loss, and the bottom panels depict the evolution of neuronal gain across training iterations. Curves represent the mean ± standard error (shaded area) over five runs with different random seeds. Dots illustrate the min-ACC per task. Color coding denotes the different optimizers: our method (NGM-SGD, Algorithm 1) in orange, momentum-SGD (MSGD) in blue, Adam in green, and vanilla SGD in gray. Plots have been zoomed in to better highlight the stability gaps. As a result, some performance drops may appear truncated.

already captured in the first task, and the subsequent tasks do not introduce additional uncertainty. Thus, the long-term gain level serves as a proxy for accumulated task complexity, echoing how biological systems adjust their internal gain to reflect changes in cognitive demand (Ferguson & Cardin, 2020; Aston-Jones & Cohen, 2005).

## 5 DISCUSSION

A foundational contribution of our work is the paradigm shift in perspective from *what* to optimize in CL to *how* to optimize it (Hess et al., 2023). Inspired by noradrenergic phasic bursts that enhance cortical responsiveness during uncertainty (Jordan, 2024; Shine, 2019), we propose gain modulation as a biologically-inspired mechanism that reframes optimization. Rather than following fixed learning trajectories in a static landscape, uncertainty-driven gain boosts dynamically reshape learning dynamics, enabling rapid adaptation and long-term retention with minimal interference. Our mechanism has two key effects: *(i)* transient gain boosts create emergent fast and slow timescales of weight adaptation (Hinton & Plaut, 1987), and *(ii)* they induce a forward-pass reparameterization in which gain increases at task switches effectively flatten the loss landscape (Dinh et al., 2017), reducing test loss and interference in a manner consistent with theoretical neuroscience models (Wainstein et al., 2025; Munn et al., 2021). While biological gain modulation is implemented through changes in spiking activity (Munn et al., 2023; Whyte et al., 2023), we show that the same principle can be instantiated in standard multilayer perceptrons (MLPs) by scaling each neuron's incoming weights with a gain coefficient, making the mechanism compatible with conventional ANN architectures.

Furthermore, rather than assuming continual learning must follow a monotonically decreasing loss path (Hess et al., 2023; Kamath et al., 2024), this approach highlights that reshaping the geometry of the landscape can offer a more flexible route to stability. Although our approach does not yield raw *overall* performance gains as hypothesized by (Hess et al., 2023), the attenuation of stability gaps it provides is crucial in real-world continual learning, where inference often occurs before training has converged and transient instabilities directly affect reliability.

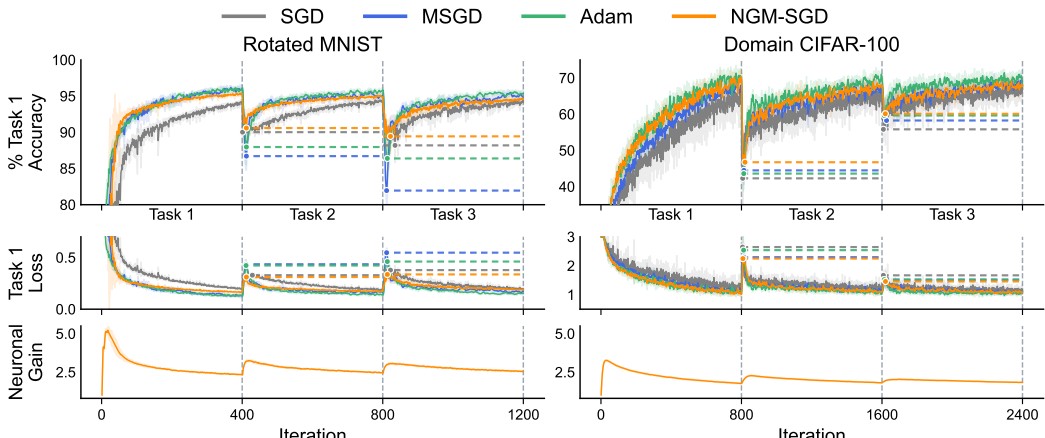

Figure 4: **Stability gaps under domain-incremental learning.** The left column shows results on Rotated MNIST with $80°$ rotations, and the right column on Domain CIFAR-100. The top panels display the test accuracy on the first task as the model is incrementally trained on all benchmark tasks. The middle panels show the corresponding test loss, and the bottom panels depict the evolution of neuronal gain across training iterations. Curves represent the mean $\pm$ standard error (shaded area) over five runs with different random seeds. Dots illustrate the min-ACC per task. Color coding denotes the different optimizers: our method (NGM-SGD, Algorithm 1) in orange, momentum-SGD (MSGD) in blue, Adam in green, and vanilla SGD in gray. Plots have been zoomed in to better highlight the stability gaps. As a result, some performance drops may appear truncated.

**Limitations and future work.** We acknowledge that our experiments rely on MNIST and CIFAR and evaluate performance over a relatively small number of tasks, with mini-ImageNet included as a more challenging benchmark, but likewise assessed over a few tasks. However, this design is intentional. Using a small task set ensures that each task induces a substantial distributional shift, making the stability gap large enough to measure reliably, whereas adding many tasks would diminish these shifts and obscure the very phenomenon we aim to study. Furthermore, the joint-training setup isolates the transient, optimization-driven forgetting that defines the stability gap, preventing confounds from other forms of interference present in more realistic CL settings. Within this controlled framework, we can predict our method's effectiveness in reducing the stability gap on more complex datasets, considering that the entropy-driven gain scaling ($\eta$) at task boundaries can be tuned according to the complexity of the dataset considered (see Figure 8A in the Appendix F).

Evaluating NGM-SGD in more realistic continual learning settings would be valuable, for example, by incorporating partial replay. Importantly, the additional experiments included in the Appendix H (Table 6) show that our method continues to attenuate the stability gap even when replay is used, confirming that the benefits of the gain dynamics persist beyond the idealized joint training setup. However, a particularly promising direction is to explore how cholinergic neuromodulation, which is central for attention and flexible learning, could support task segregation (Shine, 2019). This could be implemented through gain-gating mechanisms that temporarily silence subsets of neurons and block their updates in a sparse setting, helping to regularize learning and reduce catastrophic forgetting. Combined with transient gain boosts, such mechanisms could guide gradient updates more effectively and enable more robust *online* CL.

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

APPENDIX

The Appendix contains additional information for the content presented in the main body.

**Reproducibility and code availability.** All benchmarks are described in the Methods section of the manuscript. To enable the reproduction of all experiments, the relevant code will be made openly accessible. However, for the review process, it is provided in the zipped file.

**Computer resources.** All experiments were run on a system with an Intel Core i9-14900K CPU (8 P-cores @ 3.2–6.0 GHz, 16 E-cores @ 2.4–4.4 GHz; 32 threads), 64 GB DDR5-4800 MHz RAM (2×32 GB, CL40), and a PNY NVIDIA RTX 4000 Ada Lovelace GPU (20 GB GDDR6). Code was implemented using Python 3.10.16 in PyTorch 2.5.1 with CUDA 11.8.

**Code of ethics.** We used only publicly available datasets, and our work is confined to basic research without direct social impact. The authors declare no conflicts of interest.

## A  EVALUATION METRICS

Throughout this work, we quantify model performance using **classification accuracy** – the percentage of correctly predicted labels on a test set. Formally, for the testing dataset $D$ and model $F_\theta$, we denote the classification accuracy $\mathbf{A}(D, F_\theta)$ as the percentage of samples in $D$ that are correctly classified by $F_\theta$. In practice, we evaluate on the full test set for the first task, although a reduced evaluation set of 1,000 examples per task, sampled uniformly at random in each run, has been shown to closely approximate the full-set results (De Lange et al., 2022; Hess et al., 2023).

**Final average accuracy.** We further define the final average accuracy as the classification accuracy averaged over all contexts at the end of training. Formally, consider a continual-learning sequence of $K$ contexts (tasks). Let $F_{\theta_\mathrm{f}}^{(k)}$ denote the model parameters immediately after training on context $k$, and let $D_k$ be its corresponding held-out test set. Hence, we define

$$\textbf{avg-ACC} = \frac{1}{K} \sum_{k=1}^{K} \mathbf{A}(D_k, F_{\theta_\mathrm{f}}^{(k)}).$$

This metric captures overall end-of-sequence performance but does not reflect transient drops in accuracy that may occur during intermediate stages of training.

**Average minimum accuracy.** To assess worst-case retention within each context, we track the lowest classification accuracies observed for each context and define the average minimum accuracy as the average of these minimum accuracies. Formally, with $I_\mathcal{C}$ be the set of training-iteration indices following completion of context $k$, we have

$$\textbf{avg-min-ACC} = \frac{1}{K-1} \sum_{k=2}^{K} \min_{i \in I_\mathcal{C}} \mathbf{A}(D_k, F_{\theta_i}^{(k)}),$$

where $F_{\theta_i}^{(k)}$ indicates the model at the $i$-th training iteration in context $k$.

**Worst-case accuracy.** To capture the balance between plasticity (learning the most recent task) and stability (preserving earlier tasks) (Duong et al., 2023b; Łapacz et al., 2024), we define worst-case accuracy as

$$\textbf{WC-ACC} = \frac{1}{K} \mathbf{A}(D_k, F_{\theta_\mathrm{f}}^{(k)}) + \left(1 - \frac{1}{K}\right) \textbf{avg-min-ACC}.$$

Hence, a high **WC-ACC** indicates both effective adaptation to new tasks and minimal forgetting of prior tasks. Note that we are evaluating this at the end of the experiment, but it can also be considered at each iteration.

Table 2: Stability gaps per task for all benchmarks under joint training. Highlighted values indicate the best results.

| | | $\mathbf{SG}(T_1 \to T_2)$ | $\mathbf{SG}(T_2 \to T_3)$ | $\mathbf{SG}(T_3 \to T_4)$ | $\mathbf{SG}(T_4 \to T_5)$ |
|---|---|---|---|---|---|
| Split MNIST | NGM-SGD (ours) | $\mathbf{0.028 \pm 0.015}$ | $0.012 \pm 0.006$ | $0.014 \pm 0.008$ | $\mathbf{0.013 \pm 0.004}$ |
| | MSGD | $1.000 \pm 0.000$ | $0.032 \pm 0.013$ | $0.018 \pm 0.006$ | $0.019 \pm 0.009$ |
| | Adam | $0.883 \pm 0.133$ | $0.172 \pm 0.062$ | $0.052 \pm 0.032$ | $0.025 \pm 0.013$ |
| | SGD | $0.100 \pm 0.115$ | $\mathbf{0.009 \pm 0.005}$ | $\mathbf{0.007 \pm 0.006}$ | $0.019 \pm 0.012$ |
| Split CIFAR-10 | NGM-SGD (ours) | $\mathbf{0.133 \pm 0.097}$ | $\mathbf{0.072 \pm 0.060}$ | $\mathbf{0.072 \pm 0.094}$ | $\mathbf{0.260 \pm 0.090}$ |
| | MSGD | $0.476 \pm 0.007$ | $0.321 \pm 0.089$ | $0.116 \pm 0.117$ | $0.786 \pm 0.095$ |
| | Adam | $0.357 \pm 0.103$ | $0.148 \pm 0.222$ | $0.074 \pm 0.054$ | $0.661 \pm 0.246$ |
| | SGD | $0.389 \pm 0.037$ | $0.128 \pm 0.112$ | $0.120 \pm 0.091$ | $0.327 \pm 0.184$ |
| Split mini-ImageNet | NGM-SGD (ours) | $\mathbf{0.515 \pm 0.078}$ | $0.343 \pm 0.088$ | $\mathbf{0.213 \pm 0.162}$ | $\mathbf{0.128 \pm 0.202}$ |
| | MSGD | $0.786 \pm 0.029$ | $\mathbf{0.310 \pm 0.204}$ | $0.320 \pm 0.101$ | $0.221 \pm 0.141$ |
| | Adam | $0.587 \pm 0.133$ | $0.507 \pm 0.214$ | $0.284 \pm 0.093$ | $0.157 \pm 0.156$ |
| | SGD | $0.636 \pm 0.110$ | $0.305 \pm 0.259$ | $0.245 \pm 0.169$ | $0.141 \pm 0.212$ |
| Rotated MNIST | NGM-SGD (ours) | $0.050 \pm 0.017$ | $0.057 \pm 0.025$ | - | - |
| | MSGD | $0.097 \pm 0.021$ | $0.138 \pm 0.020$ | - | - |
| | Adam | $0.085 \pm 0.020$ | $0.099 \pm 0.010$ | - | - |
| | SGD | $\mathbf{0.045 \pm 0.033}$ | $\mathbf{0.062 \pm 0.026}$ | - | - |
| Domain CIFAR-100 | NGM-SGD (ours) | $0.334 \pm 0.070$ | $\mathbf{0.114 \pm 0.047}$ | - | - |
| | MSGD | $\mathbf{0.329 \pm 0.063}$ | $0.135 \pm 0.038$ | - | - |
| | Adam | $0.370 \pm 0.062$ | $0.142 \pm 0.049$ | - | - |
| | SGD | $0.349 \pm 0.087$ | $0.137 \pm 0.057$ | - | - |

**Average stability gap.** We define the average stability gap as the mean relative drop in classification accuracy that occurs when shifting from one context to the next (Łapacz et al., 2024). For a sequence of $K$ contexts, the stability gap at the transition from context $k-1$ to context $k$ is given by (Table 2)

$$\mathbf{SG}_k = \frac{\mathbf{A}(D_{k-1}, F_{\theta_{\mathrm{f}}}^{(k-1)}) - \min_{i \in I_C} \mathbf{A}(D_k, F_{\theta_i}^{(k)})}{\mathbf{A}(D_{k-1}, F_{\theta_{\mathrm{f}}}^{(k-1)})},$$

and the average stability gap is then:

$$\mathbf{avg\text{-}SG} = \frac{1}{K-1} \sum_{k=2}^{K} \mathbf{SG}_k.$$

By normalizing each drop by the accuracy just before the transition, we ensure that stability gaps are directly comparable even when the baseline performances differ across tasks. A large **avg-SG** signals that the model experiences significant forgetting immediately after each context switch, underscoring its lack of robustness in continual learning settings.

## B  GAIN REPARAMETERIZATION AND LANDSCAPE FLATTENING

Let $L : \mathbb{R}^n \to \mathbb{R}$ be a twice continuous differentiable ($\mathcal{C}^2$) loss function defined over an effective parameter space $W_{\mathrm{eff}} \subseteq \mathbb{R}^n$. We introduce a reparameterization map

$$\Phi : \mathbb{R}^n \to \mathbb{R}^n, \text{ s.t. } \Phi(W) = GW$$

where $G \in \mathbb{R}^{n \times n}$ is an invertible matrix, which in the simplest case just corresponds to $G = g\mathbb{I}_n$ with scalar $g \geq 1$. We then define

$$\tilde{L} : \mathbb{R}^n \to \mathbb{R}, \text{ s.t. } \tilde{L}(W) = L(\Phi(W)) = L(GW)$$

Here, $W$ is the base parameter and $W_{\mathrm{eff}} = \Phi(W)$ is the effective parameter (used in the forward pass). We then assume: (i) $L$ is $\mathcal{C}^2$, (ii) there exists $W_{\mathrm{eff}}^\star$ with $\nabla_{W_{\mathrm{eff}}}(W_{\mathrm{eff}}^\star) = 0$, and (iii) $\nabla^2_{W_{\mathrm{eff}}}(W_{\mathrm{eff}}^\star)$ is positive definite. Since $\Phi$ is invertible, the corresponding base critical point is

$$W^\star = \Phi^{-1}(W_{\mathrm{eff}}^\star) = G^{-1}W_{\mathrm{eff}}^\star$$

**Lemma 1 (Gradient transformation):** $\forall W,\ \nabla_W \tilde{L}(W) = G^\top \nabla_{W_{\text{eff}}} L(\Phi(W))$, and in particular if $G \in \mathbb{R}^{n \times n}$ we have

$$\nabla_W \tilde{L}(W) = g \nabla_{W_{\text{eff}}} L(W_{\text{eff}}).$$

*Proof:* follows from the chain rule.

**Lemma 2 (Hessian congruence):** $\forall W,\ \nabla^2_W \tilde{L}(W) = G^\top \nabla^2_{W_{\text{eff}}} L(\Phi(W)) G$, and in particular if $G \in \mathbb{R}^{n \times n}$ we have

$$H_W(W) = g^2 H_{W_{\text{eff}}}(W_{\text{eff}}).$$

where $H_W = \nabla^2_W \tilde{L}(W)$ and $H_{W_{\text{eff}}} = \nabla^2_{W_{\text{eff}}} L$.
*Proof:* second-order chain rule for linear reparameterization.

**Corollary 1 (Eigenvalue rescaling):** Let $\{\lambda_i(W^\star)\}_{i=1}^n$ be the eigenvalues of $H_W(W^\star)$ and $\{\lambda_i^{\text{eff}}(W_{\text{eff}}^\star)\}_{i=1}^n$ those of $H_{W_{\text{eff}}}(W_{\text{eff}}^\star)$. Then, $\lambda_i^{\text{eff}} = \lambda_i(W^\star)/\sigma_i(G)^2$, where $\sigma_i(G)$ are the singular values of $G$. In the isotropic case $G = g \mathbb{I}_n$,

$$\lambda_i^{\text{eff}} = \frac{1}{g^2} \lambda_i(W^\star).$$

Hence, increasing $g$ flattens the energy landscape in the effective parameter space.
*Proof:* follows from eigen-structure under congruence with $G$.

## C  GAIN ABLATION

To further consider the effects of the NGM-SGD mechanism, we ablate the gain modulation on the MNIST benchmark (Figure 5). At first sight, the ablation appears to reduce the stability gap, since performance on previously acquired tasks shows little disruption when new tasks are introduced. However, a closer inspection reveals that this comes at the cost of severely slowed knowledge acquisition. In the absence of dynamic gain updates, the effective weight update reduces to the slow component $w_{ij}^{(\text{slow})} = g_{0,i}\, w_{ij}$, meaning that only the slow timescale contributes to learning. As a result, the system adapts poorly to new tasks, giving the misleading impression of enhanced stability. This highlights that the apparent reduction in the stability gap does not reflect improved plasticity-stability balance, but rather a failure of the network to engage in fast adaptation. The contrast with NGM-SGD underscores the importance of fast gain modulation: by rapidly amplifying updates at task boundaries, it enables integration while mitigating the stability gap on the task switch.

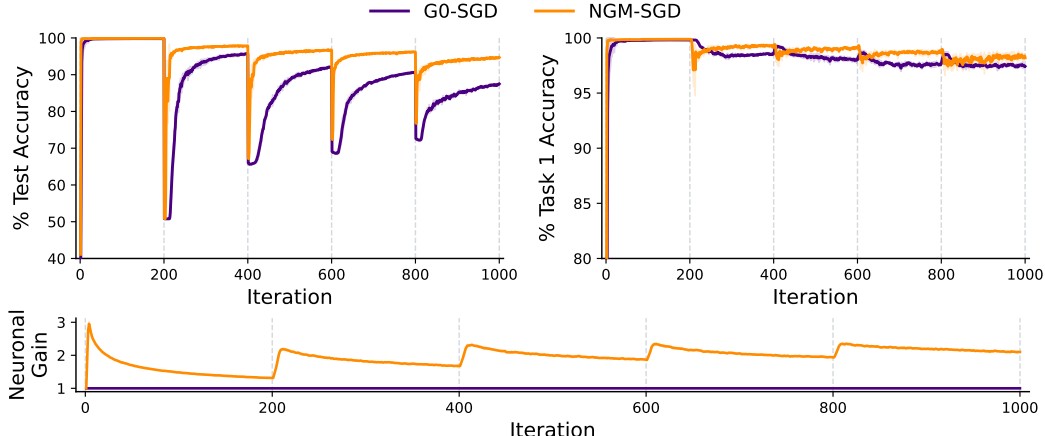

Figure 5: **Effect of ablating gain modulation on Split MNIST.** Test accuracy (top left), task-1 accuracy (top right), and neuronal gain dynamics (bottom) are shown for NGM-SGD (orange) and its ablation G0-SGD (purple).

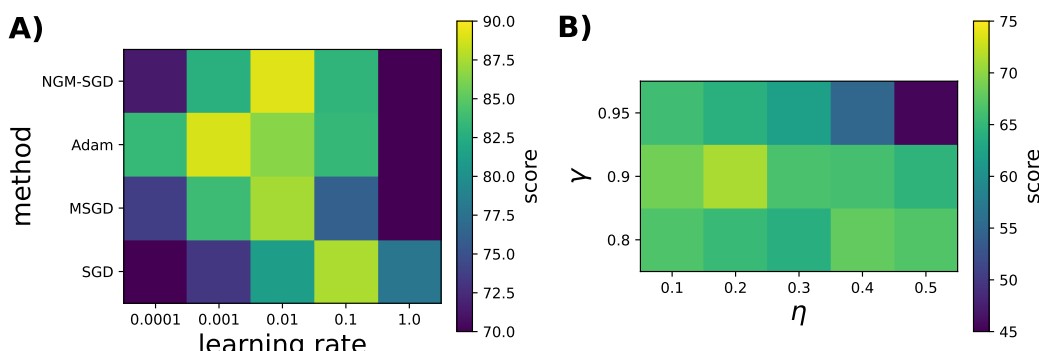

Figure 6: **Hyperparameter sweep on Split CIFAR-10. A)** Scores on best accuracy with the learning rate sweep for all methods. **B)** Scores on best accuracy and least stability gap with the hyperparameter sweep of $(\gamma, \eta)$ over the NGM-SGD method.

Table 3: Final accuracies (ACC) for all tasks in the Split CIFAR-10 benchmark comparing different learning rates.

| LR | | T1 | T2 | T3 | T4 | T5 |
|---|---|---|---|---|---|---|
| | NGM-SGD (ours) | $83.12 \pm 1.35$ | $74.14 \pm 1.75$ | $75.68 \pm 2.38$ | $75.83 \pm 2.17$ | $54.95 \pm 5.02$ |
| 1e-4 | MSGD | $80.40 \pm 1.09$ | $74.89 \pm 0.74$ | $77.15 \pm 1.00$ | $77.33 \pm 0.67$ | $61.42 \pm 1.78$ |
| | Adam | $92.07 \pm 0.67$ | $84.16 \pm 1.89$ | $87.53 \pm 0.87$ | $84.55 \pm 2.65$ | $75.02 \pm 5.24$ |
| | SGD | $55.99 \pm 4.80$ | $64.75 \pm 2.75$ | $65.12 \pm 1.73$ | $60.76 \pm 1.95$ | $55.75 \pm 2.35$ |
| | NGM-SGD (ours) | $91.09 \pm 1.46$ | $85.89 \pm 2.82$ | $85.84 \pm 1.50$ | $85.77 \pm 1.44$ | $70.84 \pm 4.72$ |
| 1e-3 | MSGD | $91.36 \pm 1.35$ | $85.31 \pm 1.77$ | $85.76 \pm 1.22$ | $86.24 \pm 1.12$ | $73.72 \pm 2.82$ |
| | Adam | $96.25 \pm 0.64$ | $91.10 \pm 5.16$ | $90.91 \pm 2.40$ | $88.79 \pm 3.09$ | $84.66 \pm 4.48$ |
| | SGD | $80.43 \pm 0.91$ | $73.61 \pm 1.62$ | $76.84 \pm 1.06$ | $77.55 \pm 0.87$ | $60.59 \pm 1.45$ |
| | NGM-SGD (ours) | $95.95 \pm 0.60$ | $91.28 \pm 2.09$ | $91.65 \pm 1.11$ | $87.81 \pm 3.99$ | $86.46 \pm 6.34$ |
| 1e-2 | MSGD | $96.31 \pm 0.45$ | $91.05 \pm 2.63$ | $90.55 \pm 2.12$ | $88.83 \pm 5.95$ | $80.43 \pm 8.17$ |
| | Adam | $93.32 \pm 1.38$ | $84.87 \pm 4.67$ | $88.99 \pm 3.07$ | $88.65 \pm 5.37$ | $84.46 \pm 2.38$ |
| | SGD | $89.95 \pm 1.62$ | $86.44 \pm 2.91$ | $85.64 \pm 1.70$ | $86.46 \pm 1.40$ | $64.28 \pm 5.37$ |
| | NGM-SGD (ours) | $90.71 \pm 1.69$ | $83.58 \pm 3.40$ | $83.88 \pm 2.60$ | $84.52 \pm 5.18$ | $81.02 \pm 4.46$ |
| 1e-1 | MSGD | $93.68 \pm 2.50$ | $79.86 \pm 4.30$ | $80.75 \pm 5.71$ | $82.93 \pm 4.75$ | $61.79 \pm 16.02$ |
| | Adam | $90.77 \pm 2.10$ | $85.28 \pm 5.04$ | $85.70 \pm 5.31$ | $89.58 \pm 2.30$ | $76.65 \pm 8.16$ |
| | SGD | $94.45 \pm 1.60$ | $89.42 \pm 3.78$ | $90.97 \pm 4.94$ | $88.06 \pm 6.46$ | $84.49 \pm 2.84$ |
| | NGM-SGD (ours) | $61.03 \pm 12.49$ | $54.06 \pm 5.41$ | $52.60 \pm 9.35$ | $51.25 \pm 12.23$ | $46.09 \pm 8.87$ |
| 1 | MSGD | $79.44 \pm 3.71$ | $58.78 \pm 12.59$ | $56.10 \pm 12.79$ | $56.02 \pm 11.34$ | $53.52 \pm 10.25$ |
| | Adam | $82.69 \pm 3.64$ | $53.35 \pm 11.77$ | $46.58 \pm 9.12$ | $50.49 \pm 0.98$ | $40.96 \pm 18.08$ |
| | SGD | $88.02 \pm 1.65$ | $80.58 \pm 4.27$ | $81.15 \pm 3.19$ | $83.17 \pm 2.11$ | $71.51 \pm 14.14$ |

# D   HYPERPARAMETER SEARCH

We conducted a systematic hyperparameter search using CIFAR-10 as a representative middle-ground benchmark. CIFAR-10 strikes a balance between easier datasets such as MNIST and more challenging ones like CIFAR-100 and mini-ImageNet, making it an appropriate testbed for controlled comparisons. For all optimizers under consideration, we performed a learning rate sweep over $\{1.0, 0.1, 0.01, 0.001, 0.0001\}$. In addition, for NGM-SGD we searched over modulation parameters, exploring $\gamma \in \{0.8, 0.9, 0.95\}$ and $\eta \in \{0.1, 0.2, 0.3, 0.4, 0.5\}$. We note that excessively high gains are both biologically implausible (Munn et al., 2023), as they lead to saturation, and practically undesirable, as they introduce instability into training.

**Learning rate search**   To identify the most effective learning rates, we evaluated end-task performance (Table 3) using a scoring criterion that rewards accuracy. Specifically, we considered the

Table 4: Average final accuracies (avg-ACC) and average stability gaps (avg-SG) in the Split CIFAR-10 benchmark comparing different combinations of the $(\gamma, \eta)$ parameters in the NGM-SGD method.

| $\gamma$ | $\eta$ | **avg-ACC** ($\uparrow$) | **avg-SG** ($\downarrow$) |
|------|------|------|------|
| 0.80 | 0.1 | $90.134 \pm 1.292$ | $0.182 \pm 0.038$ |
| 0.80 | 0.2 | $90.920 \pm 1.396$ | $0.182 \pm 0.062$ |
| 0.80 | 0.3 | $91.678 \pm 0.875$ | $0.215 \pm 0.053$ |
| 0.80 | 0.4 | $91.220 \pm 1.068$ | $0.176 \pm 0.047$ |
| 0.80 | 0.5 | $90.198 \pm 1.471$ | $0.170 \pm 0.049$ |
| 0.90 | 0.1 | $91.098 \pm 1.263$ | $0.159 \pm 0.054$ |
| 0.90 | 0.2 | $90.630 \pm 1.576$ | $0.134 \pm 0.044$ |
| 0.90 | 0.3 | $90.550 \pm 1.751$ | $0.171 \pm 0.053$ |
| 0.90 | 0.4 | $90.476 \pm 0.848$ | $0.178 \pm 0.057$ |
| 0.90 | 0.5 | $90.156 \pm 1.256$ | $0.201 \pm 0.043$ |
| 0.95 | 0.1 | $90.094 \pm 0.966$ | $0.191 \pm 0.043$ |
| 0.95 | 0.2 | $89.516 \pm 1.728$ | $0.190 \pm 0.047$ |
| 0.95 | 0.3 | $91.156 \pm 1.190$ | $0.242 \pm 0.038$ |
| 0.95 | 0.4 | $90.846 \pm 1.186$ | $0.237 \pm 0.110$ |
| 0.95 | 0.5 | $89.042 \pm 1.676$ | $0.325 \pm 0.094$ |

effective accuracy, defined as the average accuracy penalized by its variability, which favors configurations that achieve both high performance and robustness (score$_{\text{LR}}$ = avg_acc $-$ std_acc). This analysis revealed that the best values are $0.1$ for SGD, $0.01$ for MSGD and NGM-SGD, and $0.001$ for Adam (Figure 6a). These settings consistently yielded the best performance across datasets, except Rotated-MNIST, where MSGD achieved higher accuracy with a learning rate of $0.1$.

**NGM-SGD hyperparameters**  The NGM-SGD method presents three parameters: the gain baseline $g_0$, which we set to $1$ since its effect can be absorbed into the learning rate, the decay $\gamma$, and the phasic bump $\eta$. To explore which parameter values fit best for CIFAR-10, we computed a score that jointly accounts for accuracy and stability (Table 4). In particular, we considered an effective accuracy that penalizes variability, and a stability gap term that penalizes both its magnitude and variability (score$_{\text{NGM}}$ = (avg_acc $-$ std_acc) $- 100 \cdot$ (avg_sg $+$ std_sg)). The best-performing combination was $(\gamma, \eta) = (0.9, 0.2)$ (Figure 6b). While the choice of $\gamma = 0.9$ has proven consistently robust across all other tasks, the value of $\eta$ requires tuning depending on task difficulty. Higher values of $\eta$ can be used for easier datasets, whereas lower values are preferable for more complex datasets to avoid instabilities when uncertainty is higher.

## E    GAIN-UNCERTAINTY AND TASK COMPLEXITY

To explore whether neuronal gain reflects task complexity, we computed the mean gain across the first three tasks in both class-incremental and domain-incremental settings and fitted a linear trend (Figure 7). Since in class-incremental scenarios the novelty added by Tasks 4 and 5 is relatively smaller, we focus on the first three tasks, where changes in complexity are most pronounced.

As shown in Figure 7, the slopes for Split MNIST, Split CIFAR-10, and Split mini-ImageNet are positive, consistent with the idea that increased task complexity drives up sustained gain. By contrast, Rotated MNIST and Domain CIFAR-100 exhibit slightly negative slopes, indicating no increase in complexity across tasks. These patterns suggest that neuronal gain may serve as a proxy for task uncertainty and complexity, in line with evidence that biological systems adjust internal gain to match changing cognitive demands (Aston-Jones & Cohen, 2005; Ferguson & Cardin, 2020).

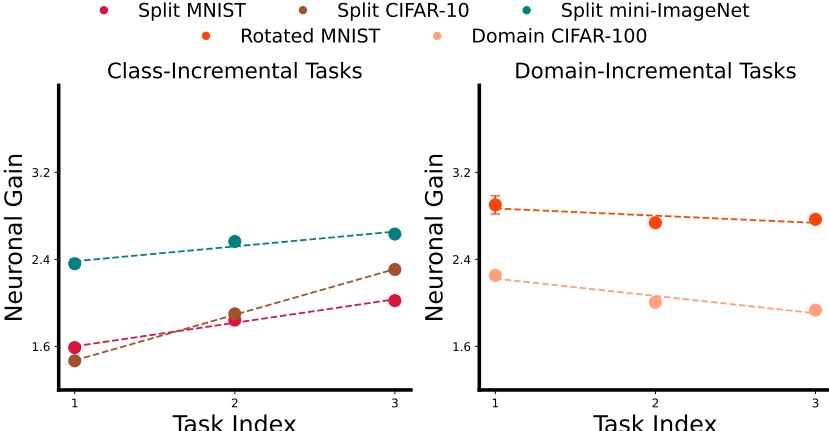

Figure 7: **Gain across tasks.** Plots of the mean neuronal gain in the first three tasks for class-incremental (left) and domain-incremental (right) experiments. We used a linear fit and obtained the slopes $2.16 \times 10^{-1}$ for Split MNIST, $4.20 \times 10^{-1}$ for Split CIFAR-10, $1.36 \times 10^{-1}$ for Split mini-ImageNet, $-6.63 \times 10^{-2}$ for Rotated MNIST, and $-1.59 \times 10^{-1}$ for Domain CIFAR-100.

## F  GAIN DYNAMICS

In the main text, we model task-dependent modulation of plasticity through a scalar gain $g_i(t)$ that multiplicatively scales the incoming weights of each output neuron $i$. The gain evolves according to the discrete-time update provided by Equation 4 where $g_0$ is a basal gain level, $H(y_t)$ is the entropy of the model's output at time $t$, $\gamma \in (0, 1)$ controls the decay of the gain towards baseline, and $\eta > 0$ scales the impact of the uncertainty signal. This dynamics is motivated by experimental findings showing that noradrenergic arousal reports surprise and modulates neuronal gain during perceptual updating (Wainstein et al., 2025; Jordan, 2024). In particular, Wainstein et al. (2025) model phasic gain increases and their subsequent tonic decay using a first-order linear differential equation driven by uncertainty, a structure closely mirrored by our update rule. In what follows, we derive the corresponding continuous-time form to make this connection explicit, and then clarify how the discrete-time equation relates to a classical exponential moving average (EMA) – a formulation more familiar in computing contexts.

### F.1  CONTINUOUS-TIME GAIN

We will derive the continuous-time form of the gain update starting from Equation 4. First, we subtract $g(t)$ from both sides, i.e.

$$g(t + 1) - g(t) = \gamma \, g(t) + (1 - \gamma)g_0 + \eta \, H(y) - g(t)$$
$$= (\gamma - 1) \, g(t) + (1 - \gamma)g_0 + \eta \, H(y)$$
$$= (1 - \gamma) \, (g_0 - g(t)) + \eta \, H(y).$$

Next, dividing both sides by the discrete time interval $\Delta t$, we obtain

$$\frac{g(t + 1) - g(t)}{\Delta t} = \frac{(1 - \gamma)}{\Delta t} \, (g_0 - g(t)) + \frac{\eta}{\Delta t} H(y).$$

Considering Euler's method for numerical integration and defining the characteristic time constant $\tau = \Delta t / (1 - \gamma)$, and the coefficient $\kappa = (\eta / \Delta t) \, \tau = \eta / (1 - \gamma)$, we arrive at the continuous-time differential equation

$$\tau \frac{dg(t)}{dt} = (g_0 - g(t)) + \kappa \, H(y) \tag{6}$$

This continuous-time formulation makes explicit that $g(t)$ behaves as a low-pass filter of the entropy signal $H(y_t)$ around the basal gain level $g_0$, with time constant $\tau$. Transient increases in uncertainty drive $g(t)$ above baseline, followed by a gradual relaxation back to $g_0$, reminiscent of noradrenergic bursts and their subsequent decay in arousal-related signals, such as pupil dilation, during perceptual transitions and unexpected events (Wainstein et al., 2025).

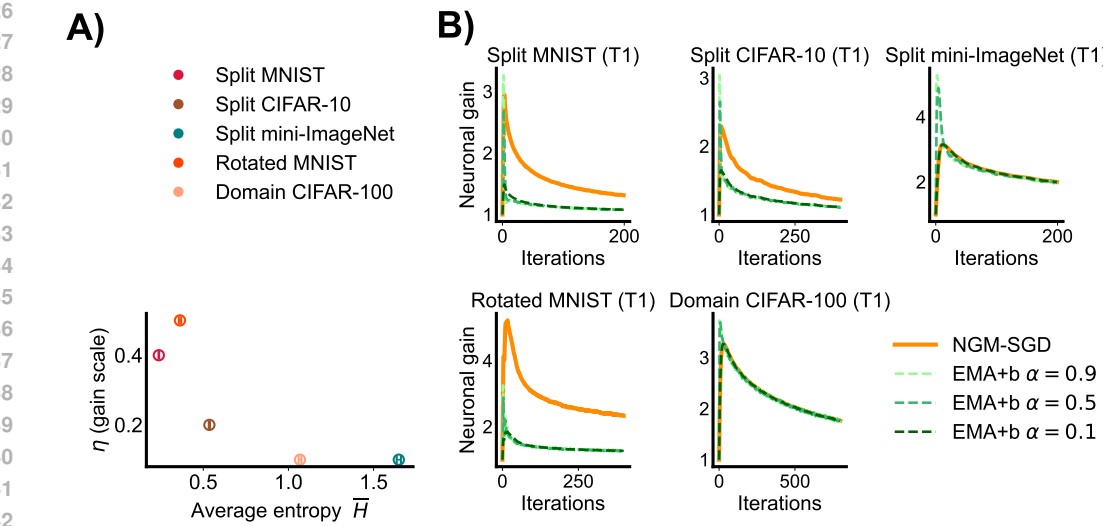

Figure 8: **A)** Relationship between the average output entropy of each dataset and the corresponding gain scale. **B)** Gain trajectories (Task 1) comparing NGM-SGD with EMA-based baselines for the domain- and class-incremental benchmarks. EMA (+ baseline) settings match only the special case $\eta = 1-\gamma$, whereas NGM-SGD allows independent control of decay and entropy-driven modulation.

### F.2 RELATION TO EXPONENTIAL MOVING AVERAGES

The discrete-time gain update in Equation 4 is reminiscent of an EMA conditioned on entropy. Here, we make this connection explicit and clarify the differences with a standard EMA. A canonical EMA of a signal $x_t$ around an equilibrium level $g_0$ is typically written as

$$g_{\mathrm{EMA}}(t+1) = (1-\alpha)\, g_{\mathrm{EMA}}(t) + \alpha\big[g_0 + x_t\big], \qquad \alpha \in (0,1), \tag{7}$$

where $\alpha \in (0,1)$ sets the smoothing factor in the exponential moving average. If we choose $x_t = H(y_t)$ and expand Equation 7, we obtain

$$g_{\mathrm{EMA}}(t+1) = (1-\alpha)\, g_{\mathrm{EMA}}(t) + \alpha g_0 + \alpha H(y_t).$$

Comparing this to the NGM-SGD update in Equation 4, shows that the two dynamics coincide if and only if

$$\gamma = 1-\alpha \quad \text{and} \quad \eta = \alpha.$$

In other words, when $\eta = 1 - \gamma$, we can rewrite our gain update exactly as an EMA of the entropy signal around the baseline $g_0$. This special case makes precise in what sense our formulation "re-calls" an EMA with a baseline term – as happens with the cases of Split mini-ImageNet and Domain CIFAR-10 (see Figure 8 B). Hence, the single parameter $\alpha$ plays a dual role: it determines both the temporal decay factor $(1-\alpha)$ and the scaling of the new signal $\alpha H(y_t)$. Increasing $\alpha$ makes the process more reactive but also less smooth; decreasing $\alpha$ increases smoothness but suppresses the impact of new information ass shown in Figure 8 B. In contrast, in Equation 4 the decay of the gain is governed solely by $\gamma$, while the influence of the entropy signal is controlled by $\eta$. This decoupling allows us, for example, to choose a large $\gamma$ (slow decay, strong stability) together with a large $\eta$ (large gain bursts at task transitions), leading to a better attenuation of the stability gap [4].

Furthermore, this decoupling is essential in practice: across benchmarks we keep $\gamma = 0.9$ to ensure slow decay and strong stability, while tuning $\eta$ to reflect the entropy structure of each dataset. Datasets with lower overall entropy require larger $\eta$ to induce sufficiently large gain transients and achieve measurable reductions in the stability gap, whereas more complex datasets with inherently higher entropy call for smaller $\eta$ to keep the gain dynamics within a bounded regime, as illustrated in Figure 8 A.

---

[4]For instance, Table 4 shows how combinations of $(\gamma, \eta)$ of EMA-equivalent settings $\{(0.9, 0.1); (0.8, 0.2)\}$ are outperformed by the case $(0.9, 0.2)$, which lies strictly outside the manifold $\eta = 1 - \gamma$.

## G  MOMENTUM RESETTING IN MULTI-TIMESCALE OPTIMIZERS

To further examine whether momentum-driven inertia contributes to the stability gap, we repeated our experiments across all benchmarks using a modified version of Adam and MSGD in which the internal momentum buffers were cleared at every task change. This was implemented by providing a task-change signal that forces the optimizer to clear its internal state, thereby removing all previously accumulated velocity – an approach that is not task-agnostic, unlike neuromodulatory gain (our NGM-SGD method). As shown in Table 5, momentum resets indeed reduce part of the instability in most of the cases (see highlighted colors), confirming that accumulated velocity can overshoot under abrupt loss-landscape shifts. However, even with perfect knowledge of task switches, these reset variants remain less stable than NGM-SGD across benchmarks.

Table 5: **Oracle reset results across benchmarks.** Main quantitative metrics for all optimizers across benchmarks under joint training with momentum resets. Bold values indicate the best result per column. For Adam and MSGD, color highlights denote the metrics in which the momentum-reset variant outperforms its non-reset counterpart (Green for Adam Reset and Blue for MSGD Reset).

| | | avg-ACC (↑) | avg-min-ACC (↑) | WC-ACC (↑) | avg-SG (↓) |
|---|---|---|---|---|---|
| | NGM-SGD (ours) | 99.015 ± 0.047 | **97.570 ± 0.465** | **97.696 ± 0.373** | **0.017 ± 0.005** |
| | SGD | 98.975 ± 0.099 | 95.887 ± 2.887 | 96.293 ± 2.311 | 0.034 ± 0.029 |
| Split | Adam | **99.088 ± 0.108** | 71.047 ± 3.753 | 76.537 ± 3.003 | 0.283 ± 0.038 |
| MNIST | Adam Reset | 99.075 ± 0.147 | 86.563 ± 6.869 | 88.921 ± 5.496 | 0.127 ± 0.069 |
| | MSGD | 98.971 ± 0.086 | 72.518 ± 0.403 | 77.628 ± 0.329 | 0.267 ± 0.004 |
| | MSGD Reset | 98.939 ± 0.100 | 72.823 ± 0.374 | 77.876 ± 0.305 | 0.264 ± 0.004 |
| | NGM-SGD (ours) | 90.630 ± 1.576 | **79.485 ± 3.875** | **80.880 ± 3.349** | **0.134 ± 0.043** |
| | SGD | 89.478 ± 1.909 | 68.712 ± 4.958 | 71.868 ± 4.007 | 0.241 ± 0.059 |
| Split | Adam | 90.342 ± 1.579 | 63.450 ± 7.839 | 67.692 ± 6.335 | 0.310 ± 0.088 |
| CIFAR-10 | Adam Reset | 90.734 ± 1.447 | 74.890 ± 4.980 | 76.302 ± 4.134 | 0.193 ± 0.054 |
| | MSGD | 89.434 ± 2.134 | 52.828 ± 3.898 | 58.348 ± 3.521 | 0.425 ± 0.044 |
| | MSGD Reset | 89.976 ± 1.175 | 70.492 ± 5.190 | 73.118 ± 4.177 | 0.230 ± 0.058 |
| | NGM-SGD (ours) | **48.312 ± 2.289** | **33.165 ± 2.435** | **36.040 ± 2.058** | **0.300 ± 0.071** |
| | SGD | 42.988 ± 2.549 | 28.345 ± 3.671 | 30.288 ± 3.309 | 0.332 ± 0.098 |
| Split | Adam | 47.208 ± 2.044 | 28.300 ± 3.371 | 32.052 ± 2.768 | 0.384 ± 0.077 |
| mini-ImageNet | Adam Reset | 46.760 ± 1.903 | 20.690 ± 4.813 | 25.264 ± 3.949 | 0.561 ± 0.106 |
| | MSGD | 46.824 ± 2.176 | 26.775 ± 2.417 | 30.416 ± 2.029 | 0.409 ± 0.067 |
| | MSGD Reset | 44.372 ± 2.678 | 28.925 ± 2.337 | 31.176 ± 2.162 | 0.335 ± 0.076 |
| | NGM-SGD (ours) | 94.948 ± 0.121 | **90.015 ± 1.451** | **91.542 ± 0.973** | 0.054 ± 0.015 |
| | SGD | 94.139 ± 0.255 | 89.107 ± 1.935 | 90.787 ± 1.300 | **0.053 ± 0.021** |
| Rotated | Adam | **95.777 ± 0.195** | 87.177 ± 1.060 | 89.910 ± 0.721 | 0.092 ± 0.011 |
| MNIST | Adam Reset | 95.545 ± 0.235 | 88.499 ± 1.073 | 90.795 ± 0.727 | 0.075 ± 0.012 |
| | MSGD | 95.297 ± 0.217 | 84.329 ± 1.359 | 87.823 ± 0.914 | 0.117 ± 0.014 |
| | MSGD Reset | 95.329 ± 0.216 | 85.902 ± 1.918 | 88.880 ± 1.282 | 0.101 ± 0.020 |
| | NGM-SGD (ours) | 68.700 ± 0.881 | **53.420 ± 2.804** | **58.290 ± 1.916** | 0.224 ± 0.042 |
| | SGD | 65.463 ± 1.204 | 49.050 ± 3.177 | 54.950 ± 2.229 | 0.243 ± 0.052 |
| Domain | Adam | 69.163 ± 0.871 | 51.615 ± 2.653 | 57.323 ± 1.840 | 0.256 ± 0.040 |
| CIFAR-100 | Adam Reset | 69.823 ± 1.111 | 49.415 ± 3.210 | 55.970 ± 2.201 | 0.295 ± 0.048 |
| | MSGD | 67.160 ± 0.994 | 51.370 ± 2.234 | 56.867 ± 1.556 | 0.232 ± 0.037 |
| | MSGD Reset | 67.463 ± 1.075 | 52.350 ± 2.248 | 57.943 ± 1.567 | **0.214 ± 0.038** |

## H  BENCHMARKS UNDER PARTIAL REPLAY

To evaluate our optimizer in a more realistic scenario, we also consider experiments in an experience replay (ER) setting (Rolnick et al., 2019). In this regime, each dataset is equipped with a finite replay buffer that is class-balanced and capped at a fixed size. After each task, we generate the buffer in a class-balanced manner, and during training, we construct each batch as a homogeneous mixture of current-task examples and replayed examples. This ER protocol ensures that the replay stream is not biased towards the most recent task, all classes remain represented, and optimizers are compared under the same rehearsal conditions.

Table 6: **Experience replay results across all benchmarks.** We report the main quantitative metrics under different replay-buffer sizes. Each method is evaluated with class-balanced replay and mixed batches of new and replayed samples. Bold values indicate the best result per column. For Adam and MSGD, color highlights denote the metrics in which the momentum-reset variant outperforms its non-reset counterpart (Green for Adam Reset and Blue for MSGD Reset).

| | | avg-ACC (↑) | avg-min-ACC (↑) | WC-ACC (↑) | avg-SG (↓) |
|---|---|---|---|---|---|
| Split MNIST (1K buffer) | NGM-SGD (ours) | **98.321 ± 0.182** | **95.123 ± 1.219** | **95.344 ± 0.988** | **0.038 ± 0.012** |
| | SGD | 97.991 ± 0.319 | 94.251 ± 1.813 | 94.517 ± 1.470 | 0.044 ± 0.018 |
| | Adam | 97.974 ± 0.259 | 48.515 ± 5.823 | 57.973 ± 4.662 | 0.505 ± 0.059 |
| | Adam Reset | 97.918 ± 0.175 | 81.849 ± 8.599 | 84.700 ± 6.880 | 0.166 ± 0.086 |
| | MSGD | 97.927 ± 0.249 | 52.712 ± 3.238 | 61.280 ± 2.600 | 0.462 ± 0.033 |
| | MSGD Reset | 97.952 ± 0.103 | 55.560 ± 3.841 | 63.597 ± 3.073 | 0.433 ± 0.039 |
| Split CIFAR-10 (1K buffer) | NGM-SGD (ours) | 62.788 ± 1.521 | **50.372 ± 2.887** | 43.170 ± 2.374 | **0.334 ± 0.044** |
| | SGD | 63.936 ± 1.798 | 40.445 ± 4.002 | 35.506 ± 3.374 | 0.483 ± 0.057 |
| | Adam | **69.820 ± 2.930** | 46.098 ± 4.573 | 42.640 ± 4.346 | 0.431 ± 0.060 |
| | Adam Reset | 64.944 ± 3.043 | 48.410 ± 4.164 | **44.334 ± 3.506** | 0.350 ± 0.076 |
| | MSGD | 65.576 ± 1.986 | 35.435 ± 5.486 | 31.834 ± 4.447 | 0.568 ± 0.075 |
| | MSGD Reset | 66.312 ± 2.893 | 39.103 ± 3.021 | 35.240 ± 2.780 | 0.522 ± 0.045 |
| Split CIFAR-10 (3K buffer) | NGM-SGD (ours) | 76.722 ± 1.795 | **63.862 ± 5.975** | **60.326 ± 4.833** | **0.248 ± 0.077** |
| | SGD | 73.652 ± 3.375 | 49.898 ± 7.246 | 47.862 ± 6.113 | 0.394 ± 0.096 |
| | Adam | 79.076 ± 2.514 | 50.398 ± 6.527 | 51.322 ± 5.559 | 0.412 ± 0.081 |
| | Adam Reset | **80.856 ± 2.313** | 50.898 ± 4.780 | 51.728 ± 3.968 | 0.422 ± 0.058 |
| | MSGD | 77.644 ± 2.921 | 34.215 ± 8.591 | 37.202 ± 7.282 | 0.613 ± 0.101 |
| | MSGD Reset | 76.974 ± 2.489 | 36.532 ± 6.987 | 39.122 ± 5.675 | 0.577 ± 0.086 |
| Split mini-ImageNet (1K buffer) | NGM-SGD (ours) | 23.760 ± 1.236 | **12.300 ± 1.442** | **11.308 ± 1.171** | 0.503 ± 0.055 |
| | SGD | 19.044 ± 2.201 | 10.345 ± 2.036 | 9.492 ± 1.740 | **0.485 ± 0.110** |
| | Adam | 23.236 ± 2.293 | 11.500 ± 1.181 | 10.824 ± 1.025 | 0.493 ± 0.115 |
| | Adam Reset | 23.344 ± 2.075 | 8.110 ± 1.559 | 8.424 ± 1.650 | 0.703 ± 0.088 |
| | MSGD | 22.160 ± 1.123 | 8.735 ± 0.963 | 8.196 ± 0.822 | 0.547 ± 0.086 |
| | MSGD Reset | 22.628 ± 1.500 | 10.345 ± 1.939 | 9.536 ± 1.589 | 0.493 ± 0.100 |
| Split mini-ImageNet (5K buffer) | NGM-SGD (ours) | **41.004 ± 2.139** | **20.435 ± 2.995** | **22.016 ± 2.517** | 0.541 ± 0.071 |
| | SGD | 38.008 ± 2.553 | 19.480 ± 3.139 | 21.608 ± 2.699 | **0.490 ± 0.090** |
| | Adam | 39.916 ± 2.115 | 15.625 ± 2.222 | 18.744 ± 1.973 | 0.621 ± 0.054 |
| | Adam Reset | 40.812 ± 2.082 | 11.295 ± 3.039 | 15.228 ± 2.737 | 0.748 ± 0.069 |
| | MSGD | 38.404 ± 1.924 | 15.865 ± 3.251 | 18.440 ± 2.669 | 0.589 ± 0.090 |
| | MSGD Reset | 38.748 ± 2.121 | 14.565 ± 1.951 | 17.284 ± 1.832 | 0.618 ± 0.054 |
| Rotated MNIST (2K buffer) | NGM-SGD (ours) | **92.974 ± 0.221** | 87.731 ± 1.722 | 88.981 ± 1.160 | **0.064 ± 0.019** |
| | SGD | 91.986 ± 0.240 | 83.096 ± 3.617 | 85.690 ± 2.421 | 0.103 ± 0.040 |
| | Adam | 92.956 ± 0.263 | 82.554 ± 1.692 | 85.415 ± 1.142 | 0.121 ± 0.019 |
| | Adam Reset | 92.655 ± 0.271 | 86.311 ± 1.033 | 87.986 ± 0.708 | 0.075 ± 0.012 |
| | MSGD | 92.597 ± 0.363 | 80.860 ± 1.307 | 84.255 ± 0.887 | 0.135 ± 0.015 |
| | MSGD Reset | 92.887 ± 0.347 | 81.000 ± 2.222 | 84.540 ± 1.494 | 0.135 ± 0.025 |
| Domain CIFAR-100 (1K buffer) | NGM-SGD (ours) | 51.220 ± 0.972 | 39.255 ± 2.888 | 40.150 ± 1.942 | 0.262 ± 0.070 |
| | SGD | 48.220 ± 1.509 | 32.165 ± 2.792 | 34.947 ± 1.930 | 0.363 ± 0.074 |
| | Adam | **53.127 ± 0.647** | 41.140 ± 4.926 | 42.423 ± 3.305 | 0.248 ± 0.083 |
| | Adam Reset | 52.743 ± 1.112 | 41.930 ± 2.188 | 42.923 ± 1.594 | **0.212 ± 0.051** |
| | MSGD | 50.150 ± 0.916 | 37.045 ± 2.477 | 38.763 ± 1.697 | 0.282 ± 0.060 |
| | MSGD Reset | 51.060 ± 1.036 | 39.205 ± 2.454 | 40.387 ± 1.707 | 0.254 ± 0.061 |
| Domain CIFAR-100 (5K buffer) | NGM-SGD (ours) | 59.927 ± 0.934 | 43.110 ± 3.269 | 46.497 ± 2.209 | 0.316 ± 0.055 |
| | SGD | 56.317 ± 0.781 | 39.660 ± 3.946 | 43.177 ± 2.647 | 0.332 ± 0.066 |
| | Adam | 60.287 ± 1.155 | 44.040 ± 3.110 | 47.767 ± 2.154 | **0.296 ± 0.053** |
| | Adam Reset | 61.577 ± 0.911 | 44.120 ± 2.349 | 47.783 ± 1.650 | 0.310 ± 0.038 |
| | MSGD | 58.373 ± 0.910 | 39.830 ± 2.043 | 44.260 ± 1.414 | 0.343 ± 0.036 |
| | MSGD Reset | 57.213 ± 1.207 | 41.370 ± 3.691 | 44.790 ± 2.513 | 0.307 ± 0.064 |

However, when the replay buffer is very limited (e.g. 1K samples), especially on harder benchmarks such as Split CIFAR-10, Split mini-ImageNet, and Domain CIFAR-100, stability gap measurements become entangled with steady-state forgetting, which can obscure the phenomenon we aim to isolate (Figure 9). With insufficient capacity to store representative coverage of past tasks, the optimizer

converges to genuinely suboptimal solutions on older tasks, leading to a long-term loss of performance that can exceed the transient drop induced by the stability gap itself. This behavior aligns with our rationale for studying the stability gap in isolation: the transient form of forgetting should be distinguished from the steady-state forgetting that arises from poor approximations of the joint loss (*catastrophic forgetting*, Figure 1). For this reason, we also ran experiments with larger buffers, enabling models to recover performance on earlier tasks after each switch and revealing the stability gap in a cleaner, more interpretable manner.

Overall, ER experiments confirm that NGM SGD continues to attenuate the stability gap even in a more realistic CL scenario where replay partially mitigates catastrophic forgetting, although in Domain CIFAR-100 Adam achieves higher performance metrics, potentially because it converges to a more favorable minimum.. Across the benchmarks reported in Table 6, our method generally shows smaller stability drops and higher minimum accuracies than the other optimizer baselines, indicating that neuromodulatory gain dynamics remain effective beyond the joint training setting. The reset variants of MSGD and Adam exhibit mixed behaviour across benchmarks, sometimes reducing stability gaps and sometimes increasing them, indicating that resetting the optimizer state at task boundaries can lead to unstable optimization under replay. Moreover, these methods rely on perfect knowledge of task boundaries and explicitly clear their internal state, so they should be viewed as oracle controls rather than realistic CL baselines.

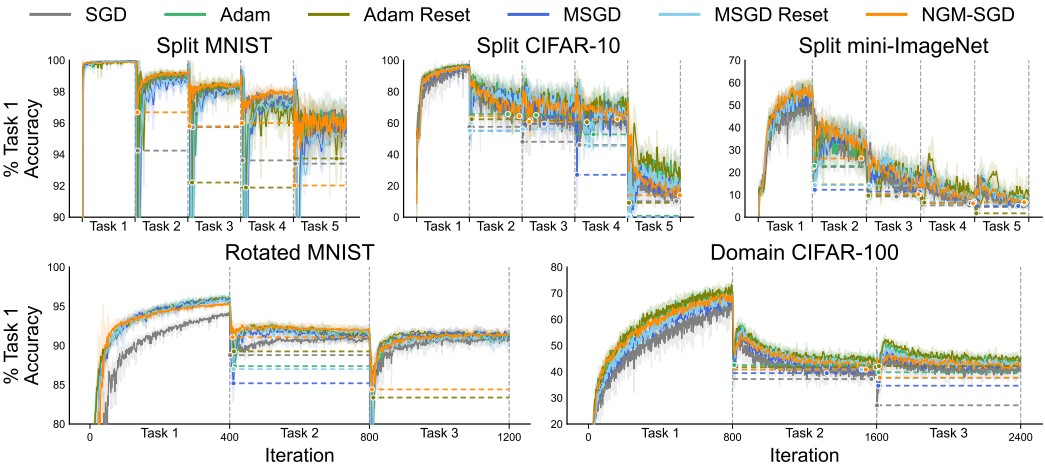

Figure 9: **Task 1 accuracy across replay experiments with a 1K buffer.** Each panel shows per-iteration accuracy at task switches for all optimizers under a limited replay memory. On simple benchmarks (Split MNIST and Rotated MNIST), the stability gap is visible, whereas on more complex datasets (Split CIFAR-10, Split mini-ImageNet, and Domain CIFAR-100), the small buffer induces substantial steady-state forgetting that dominates over the transient drop. This illustrates how poor approximations of the joint loss can obscure the stability gap we aim to study.

