# OpenReview forum: "Noradrenergic-inspired gain modulation attenuates the stability gap in joint training"
_ICLR.cc/2026/Conference — Submitted to ICLR 2026_

### Official Review · Reviewer_KNC8 · 2025-10-27

**Soundness:** 2
**Presentation:** 3
**Contribution:** 3
**Rating:** 4
**Confidence:** 4

**Summary:**

The stability gap in Continual Learning is a phenomenon in which performance temporarily declines when a new task is introduced. Previous studies have shown that this occurs mainly because the strong signal from the loss leads to rapid and strong weight modifications. The authors of this study examined this phenomenon in deep learning models from the perspective of biological models. Biological models can learn new tasks through a multi-timescale dynamics that gradually allows new information to be incorporated into neurons. Inspired by this, the authors present an optimiser that accounts for two-timescale dynamics (fast and slow), enabling the balance between adaptation and stability in sequentially learned models. To study the problem in detail, the experiments focus on an environment with sequential tasks but assuming that all previous tasks are accessible (joint learning). Experiments are conducted on different variants of MNIST, CIFAR, and Mini ImageNet.

**Strengths:**

- Works inspired by biological models help us better understand how we can expand the capabilities of current models. The authors do a good job of linking a specific limitation (the stability gap) of current models to mechanisms that can inspire future improvements.
- The idea of adapting the optimiser with a two-timescale approach is interesting. As the authors mention, it follows the line of work that seeks fast adaptations (fast weights) and longer-term adaptations.

**Weaknesses:**

- The main contribution of the work is an optimiser that approximates the two-timescale gradient descent strategy to mitigate the stability gap. However, no significant benefits are seen in the results. Table 1 shows a slight improvement over SGD but introduces a level of complexity that has not been studied. Figure 2 shows that NGM-SGD helps control the loss in Task 1, but there is no significant difference compared to SGD.
- At the beginning of Section 4, it is stated that the aim is to study ‘How to optimise and not what to optimise’, but I believe the two components are closely related and difficult to study independently.
    - Could you explain in more detail what you mean?
- The paper raises two very interesting questions in the introduction (lines 62-64). However, I am not sure that they are answered in the paper. In Section 3, the method is presented, but no direct connection is made to the biological brain.

**Questions:**

- Can we view "g" as a value similar to the momentum of other optimisers, but which changes as described in equation 2?
- Did you conduct experiments in which you reset the optimisers for each task? As mentioned in the paper, the momentum of Adam or M-SGD can negatively affect the stability gap, which can be tackled by resetting the momentum values.
- Did you conduct experiments with different values for the number of iterations? Previous work has shown that models that are trained for more iterations achieve greater stability. It would be interesting to study this statement under this scenario.
- The motivation for proposing this new optimiser is to reduce the strong weight modifications by using the g values (which should capture most of the task change). Did you study how the changes in the weights "w" compare with those of other optimisers? Could the reduction in plasticity have a long-term effect on the model's performance?
    - Conclusions may not be difficult to obtain for simple tasks or those with few iterations. More complex tasks or those with greater changes in task distribution may have different behaviours.
- Line 303 mentions that they restrict modulation to only the last layer of the ResNet. Did you experiment with the entire model? Did you conduct experiments with pre-trained models (ResNet pre-trained with ImageNet and on datasets outside the distribution, such as CUB)?
    - The latter may be interesting, as it would give an idea of how the optimiser behaves when big variations in the weights are needed.

---

> ### Author Response · Authors · 2025-11-17
>
> We thank the reviewer for the insightful feedback. Please find the responses to your questions and concerns below (Weakness W, Question Q).
>
> **W1:** We thank the reviewer for pointing this out. The original computation of avg-min-ACC and WC-ACC mistakenly included the first task, i.e., the model’s initialization before any task transition. See the general comment for updated values. NGM-SGD adds adaptive dynamics but still uses only three hyperparameters, comparable to Adam and just one more than MSGD. Test loss at task transitions is consistently lower for NGM-SGD than for SGD; we will add simple visual cues to highlight this. We attribute this improvement to the flatter effective loss landscape induced by gain reparametrization.
>
> **W2:** This distinction follows the perspective in [1]. The “what” refers to the continual learning objective itself, while the “how” refers to the optimisation dynamics introduced by the optimiser. To isolate the second one, we use the ideal joint training setting, where the model optimises the true joint objective and forgetting due to objective approximations is removed. This allows us to study the transient instabilities that arise purely from optimisation dynamics (stability gap). A similar observation was made in [2], and we will clarify this in the revised manuscript.
>
> **W3:** We acknowledge this concern. The two questions in the introduction were meant as motivation, not as biological modelling. Their purpose was to motivate a bio-inspired optimisation mechanism grounded in neuroscience findings on uncertainty-driven gain modulation [3], rather than to claim a direct correspondence with biological circuits. We will clarify this more explicitly in the revised manuscript.
>
> **Q1:** Only partly. Like momentum, the gain scales the update in the backward pass, but unlike momentum it also reparametrizes the forward weights [4], which modifies the curvature of the loss landscape over the effective weight space and produces a flatter region around task transitions when evaluated on the test set. This forward effect is what distinguishes NGM-SGD and underlies its stabilising behaviour.
>
> **Q2:** This is an interesting point. While resetting momentum could reduce the instability caused by inertia terms at task boundaries, it requires knowing exactly when a task switch occurs, which is not available in our task-agnostic and online setting. In contrast, NGM-SGD adapts naturally: the gain increases when the output entropy rises under distribution shift, providing an implicit and task-agnostic response to new tasks without requiring external task information.
>
> **Q3:** When using a large number of iterations, the model overfits the first task even with augmentation, which amplifies the loss spike at the next task switch and makes the stability gap harder to interpret. We therefore use the number of iterations that yields high accuracy on the first task without overfitting, providing a stable operating point where transient instabilities can be observed clearly.
>
> **Q4:** Thank you for this question. At task transitions, all optimisers show abrupt weight changes, but this effect is noticeably smaller under NGM-SGD. The gain-driven reparametrisation absorbs part of the distribution shift, so the weights do not need to change as sharply, improving stability at the switch. We do not expect this reduced plasticity to affect long-term performance, as the optimisation objective is unchanged and the gain increase is transient before reverting to standard SGD-like behaviour. We will clarify this in the revised manuscript.
>
> **Q5:** Applying gain modulation to the convolutional layers disrupts feature learning and batch normalization statistics, so we restrict it to the classification head being enough to reduce stability effects [5]. Using a pretrained backbone on an ood dataset is essentially a class incremental setting; in that case the output entropy would rise and trigger the gain update exactly as in our benchmarks. We therefore expect behaviour similar to the class incremental results already reported, such as Split mini-ImageNet and the distribution shift in Split CIFAR-10 illustrated in Fig. 2.
>
> **References**
>
> [1]: Hess, T., Tuytelaars, T., and van de Ven, G. M. (2023). Two Complementary Perspectives to Continual Learning: Ask Not Only What to Optimize, But Also How. arXiv:2311.04898.
>
> [2]: Kamath, S., Soutif-Cormerais, A., van de Weijer, J., and Raducanu, B. (2024). The Expanding Scope of the Stability Gap. arXiv:2406.05114.
>
> [3]: Wainstein, G. et al. (2025). Gain neuromodulation mediates task-relevant perceptual switches. eLife.
>
> [4]: Dinh, L., Pascanu, R., Bengio, S., & Bengio, Y. (2017). Sharp Minima Can Generalize For Deep Nets. ICML.
>
> [5]: Łapacz, W., Marczak, D., Szatkowski, F., and Trzciński, T. (2024). Exploring the Stability Gap in Continual Learning: The Role of the Classification Head. arXiv:2411.04723.

---

> > ### Comment · Reviewer_KNC8 · 2025-11-24
> >
> > I thank the authors for their response to my comments.
> >
> > There is agreement among reviewers that framing the method in biological terms is an oversell for the paper. The authors fail to connect the proposed method to the motivation clearly. To resolve this, a significant restructuring of the manuscript is required. Something the authors mention in their responses, but it is important to validate it.
> >
> > Concerning my second question. I agree that in a task-agnostic scenario, one does not have access to when a task changes, but, similar to the motivation to tackle the stability gap, one could use the drastic change in the distribution (similar to methods that detect distribution shifts) to reset the momentum (Or when detecting a new class). Moreover, even if this is not an option, conducting the experiments would help gain more insights into why it could work (or not).
> >
> > Concerning my first question. Thanks, it helps me better understand the method. Following line 190, the method has one "g" value per classifier row, which makes me think it behaves similarly to a moving average conditioned on the entropy value. If this is the case, how does a baseline using a moving average perform?
> >
> > After reading the answers and comments from other reviewers, I decided to keep the rating pending the updated version of the manuscript.

---

> > > ### Author Response · Authors · 2025-12-02
> > >
> > > We thank the reviewer for the constructive follow-up and are pleased that several of the concerns were clarified by our previous response. In the revised manuscript, we have substantially restructured the Introduction and early sections to provide a clearer and more accessible ML-oriented motivation for the method, including a new explainer figure (Figure 1 in the revised manuscript) that illustrates the stability gap in the optimisation trajectory, distinguishing it from the catastrophic forgetting problem, making the core phenomenon more intuitive. We now explicitly separate the problem formulation, the motivation for modulating gain, and the connection to the stability gap, ensuring that the rationale of the method is stated upfront and without biological overselling. We have also clarified the role of the gain in modifying both the forward computation and the update rule, which induces an effective reparametrization of the loss landscape. This theoretical insight is now developed more formally in lines 243-246 and demonstrated mathematically in Appendix E.
> > >
> > > Regarding the question on detecting distribution shifts and resetting momentum, we agree that this is a meaningful direction. To examine the reviewer’s suggestion directly, we **added momentum-reset experiments** for MSGD and Adam across all benchmarks (Appendix G). These experiments show that resetting velocity at distribution shifts does attenuate overshooting -- consistent with the intuition discussed in the Results section (lines 406-409) --, but even under these favourable conditions, both methods still underperform NGM-SGD (our method) and sometimes even vanilla SGD (see Table~5 in the updated manuscript). This suggests that although removing accumulated inertia helps, it is insufficient to resolve the instability caused by sharp loss directions, whereas the gain-based reparametrization provides a more fundamental mitigation.
> > >
> > > Finally, we appreciate the observation that the gain update resembles an exponential moving average (EMA). In Appendix F.2, we show that NGM-SGD reduces exactly to an EMA only in the constrained case $\eta = 1-\gamma$, where a single parameter controls both decay and responsiveness. This coupling imposes a trade-off: increasing reactivity necessarily reduces smoothness, and vice versa (see Figure 8B in the revised manuscript). In contrast, NGM-SGD decouples these roles, allowing $\gamma$ to determine slow decay for stability while $\eta$ independently modulates the influence of the entropy signal. As shown in the appendix, EMA-equivalent settings underperform because they are either overly smooth or insufficiently sensitive to task transitions, whereas NGM-SGD can exploit parameter combinations outside the EMA manifold to produce stronger gain transients and smaller stability gaps. As an example, in Split CIFAR-10, Table 4 shows that EMA-equivalent settings $( \gamma, \eta )$ of EMA-equivalent settings $\{(0.9, 0.1); (0.8, 0.2)\}$ are outperformed by the case $(0.9, 0.2)$, which lies strictly outside the manifold $\eta = 1-\gamma$.

---

### Official Review · Reviewer_gDN2 · 2025-10-31

**Soundness:** 3
**Presentation:** 2
**Contribution:** 2
**Rating:** 4
**Confidence:** 3

**Summary:**

This paper identifies and formalizes the “stability gap” in continual learning—transient performance drops at task boundaries that persist even under ideal joint training—and attributes it to optimization dynamics rather than objective choice. It proposes a biologically inspired optimizer, Noradrenergic Gain-Modulated SGD (NGM-SGD), which introduces a fast timescale, uncertainty-driven neuronal gain (triggered by output entropy) atop standard weight updates, effectively scaling the loss and flattening local curvature to curb overshoot while accelerating adaptation. The authors provide an intuitive two-timescale interpretation (fast gain × slow weights) and show that modest, head-only gain modulation suffices for deep architectures. Across class- and domain-incremental benchmarks, NGM-SGD consistently narrows the stability gap (higher min-ACC, lower avg-SG) without sacrificing final accuracy.

**Strengths:**

The work draws an interesting inspiration from noradrenaline-driven gain modulation, offering a principled lens on when and how a learner should adapt under distribution shifts in continual learning.

**Weaknesses:**

1. The paper is difficult to follow. It leans heavily on terminology from biological neural circuits without sufficient plain-language grounding or progressive intuition.
2. The method is primarily targeted at narrowing the stability gap during task transitions, but it does not directly address the central challenge of continual learning—catastrophic forgetting over long horizons. The proposed gain-modulated SGD feels like a modest variant of existing two-timescale or adaptive-step-size ideas rather than a fundamentally new optimization principle.
3. The evaluation is limited to small or mid-scale datasets. Without results on larger, more realistic settings, it is hard to assess robustness, training dynamics under real-world complexity, and computational overhead.
4. The comparisons focus on relatively dated optimizer baselines, and the reported improvements are modest. The method is not evaluated within mainstream continual learning frameworks. This raises uncertainty about extensibility and practical utility.

**Questions:**

see Weakness

---

> ### Author Response · Authors · 2025-11-17
>
> We thank the reviewer for the insightful feedback. Please find the responses to your questions and concerns below (Weakness W, Question Q).
>
> **W1:** We apologise for this. In the current version, we leaned strongly on the biological motivation to provide context, which may have made the exposition harder to follow. We will revise the presentation and reframe the discussion to place greater emphasis on the ML relevance of the method.
>
> **W2:** We acknowledge that this work does not address catastrophic forgetting, which is a separate problem from the stability gap. Our focus is on transient drops at task transitions -- a phenomenon that only recently became visible under continual evaluation [1] and is practically relevant in safety-critical scenarios or where inference happens before convergence (see [2] for a full discussion). NGM-SGD mitigates these instabilities. Importantly, the gain does more than adapt the update size: it induces a reparametrization of the forward weights [3] that flattens the effective loss landscape, a stabilising mechanism not achievable through learning-rate scaling alone and so far observed only in neuroscience [4]. We will add a brief theoretical clarification in the appendix.
>
> **W3:** We adopted similar benchmarks as the ones first positioned in the paper that first identified the stability gap [1]. Adopting these benchmarks allowed us to understand the effects of our method on the attenuation of the stability gap, and more complex datasets would hinder this investigation. The use of the simplified setting of joint training (full-replay) further helps remove other sources of forgetting and lets us focus specifically on the transient instabilities at task switches (stability gaps). Furthermore, we believe that introducing more complex datasets would not affect the effectiveness of our method, as increased dataset complexity can be addressed by using a stronger backbone architecture, and this does not impair the gain effects, which are applied only to the classification layer.
>
> **W4:** We acknowledge that we can clarify our rationale of baseline comparison and will add this more explicitly in a revised version of the manuscript. The stability gap has been identified only recently [1], suggesting that its origin lies in the optimisation procedure itself [2]. Our focus is therefore on understanding whether the optimisation dynamics contribute to this phenomenon. Because NGM-SGD acts purely at the optimiser and weight-update level, the most meaningful baselines are the standard optimisers (vanilla SGD, MSGD, Adam) that underpin all continual-learning methods. We acknowledge that other works have been proposed to target this problem, but they rely on assumptions incompatible with our setting: [5] expand the classifier at every task (increasing complexity and not being task agnostic), [6] and [7] rely on pretrained backbones or LoRA-style updates (also not task agnostic), and [8] maintains a smoothed average of the weights during task transitions (computationally heavy and not task agnostic). These approaches introduce architectural changes, external modules, or task information, whereas our method operates purely at the optimiser and weight-update level.
>
> Regarding the evaluation protocol, joint training (full-replay) is a mainstream CL framework and is used deliberately to study the stability gap in isolation. Replay- and regularization-based methods introduce additional sources of forgetting that would confound the optimisation effects we aim to analyse. Under joint training, any transient instability originates directly from the optimizer, making it the most appropriate setting to evaluate the contribution of NGM-SGD (our work) and its potential to complement forgetting-targeted CL approaches.
>
> **References**
>
> [1]: De Lange, M., van de Ven, G., and Tuytelaars, T. (2022). Continual evaluation for lifelong learning: Identifying the stability gap. arXiv:2205.13452.
>
> [2]: Hess, T., Tuytelaars, T., and van de Ven, G. M. (2023). Two Complementary Perspectives to Continual Learning: Ask Not Only What to Optimize, But Also How. arXiv:2311.04898.
>
> [3]: Dinh, L., Pascanu, R., Bengio, S., & Bengio, Y. (2017). Sharp Minima Can Generalize For Deep Nets. ICML.
>
> [4]: Wainstein, G. et al. (2025). Gain neuromodulation mediates task-relevant perceptual switches. eLife.
>
> [5]: Łapacz, W., Marczak, D., Szatkowski, F., and Trzciński, T. (2024). Exploring the Stability Gap in Continual Learning: The Role of the Classification Head. arXiv:2411.04723.
>
> [6]: Harun, M. Y., and Kanan, C. (2023). Overcoming the Stability Gap in Continual Learning. arXiv:2306.01904.
> [7]: Guo, Y., Fu, J., Zhang, H., Zhao, D., and Shen, Y. (2024). Efficient Continual Pretraining by Mitigating the Stability Gap. arXiv:2406.14833.
>
> [8]: Soutif-Cormerais, A., Carta, A., and van de Weijer, J. (2023). Improving Online Continual Learning Performance and Stability with Temporal Ensembles. arXiv:2306.16817.

---

> > ### Comment · Reviewer_gDN2 · 2025-11-27
> >
> > I would like to thank the authors for their response to my comments. The authors have addressed some of my concerns; I believe the introduction would benefit from further restructuring to improve clarity and make the paper more accessible.
> >
> > While I agree that the stability gap is an important challenge in continual learning, I still think the proposed method is a somewhat tricky and modest variant of existing optimizers. A more fundamental or theoretical solution would be valuable, as well as a more thorough validation of the approach on continual learning tasks.
> >
> > After considering both the authors' response and the feedback from other reviewers, I have decided to maintain my rating.

---

> > > ### Author Response · Authors · 2025-12-02
> > >
> > > We thank the reviewer for recognising that several of their comments were addressed, and we appreciate the additional feedback. Following the initial reviews, we substantially revised the manuscript to improve clarity and accessibility. In particular, we have substantially restructured the Introduction and early sections to provide a clearer and more accessible ML-oriented motivation for the method, including a new explainer figure (Figure 1 in the revised manuscript) that illustrates the stability gap in the optimisation trajectory, distinguishing it from the catastrophic forgetting problem, making the core phenomenon more intuitive.
> > >
> > > We also strengthened the theoretical framing of the gain reparametrization by adding a more precise intuition in the main text (lines 243-246) and a full mathematical treatment in Appendix E. The mechanism it introduces is conceptually fundamental: the gain modifies both the update rule and the forward computation, inducing an effective reparametrization of the energy landscape. This dual role flattens sharp loss directions and improves optimisation stability while remaining fully task-agnostic, since the gain evolves solely from the model’s entropy signal and does not rely on task boundaries, naturally detecting the distribution shift. This situates our method well beyond a modest variant of existing optimisers and anchors it in a principled view of how neuronal sensitivity can be adaptively tuned. A genuinely modest variant of current optimisation methods would instead correspond to resetting momentum at task switches, as suggested by reviewer **KNC8**. To examine this, we **added momentum-reset experiments** for MSGD and Adam across all benchmarks (Appendix G). These experiments show that clearing the velocity at distribution shifts does reduce overshooting -- consistent with the intuition in the Results section (lines 406-409) -- yet even under these favourable conditions (perfect knowledge of task transitions), both optimisers still underperform NGM-SGD and in some cases even vanilla SGD (Table 5). This indicates that although removing accumulated inertia helps, it is insufficient to address the instability created by sharp loss directions, whereas the gain-based reparametrization provides a more fundamental mitigation.
> > >
> > > Regarding the request for more realistic continual learning settings, we emphasise that the experimental design in the main paper is intentional. We clarified this rationale in the revised Introduction and reiterated it in the *Limitations and future work* paragraph under the Discussion section. Using a small number of tasks ensures that each switch produces a clear distributional change, making the stability gap measurable. The joint training setup further isolates the transient optimisation-driven forgetting that defines this gap, avoiding confounds from steady state forgetting introduced by limited replay or weak regularisation. However, we fully agree that evaluating NGM-SGD in more realistic continual learning settings is valuable. To address this, we **added a comprehensive set of experience replay (ER) experiments** in Appendix H. As expected, when the replay buffer is small, steady state forgetting often exceeds the transient effect, which masks the stability gap and makes it difficult to measure reliably (see Figure 9 in the revised manuscript). To uncover the gap more clearly, we also ran experiments with larger buffers that allow models to recover performance on earlier tasks; under these conditions, the stability gap becomes measurable again. Across most benchmarks, NGM SGD continues to reduce this gap more effectively than the baseline optimizers, indicating that its gain dynamics remain beneficial even under more realistic replay-based continual learning scenarios (Table 6 in the revised manuscript).

---

### Official Review · Reviewer_4gom · 2025-10-31

**Soundness:** 3
**Presentation:** 3
**Contribution:** 1
**Rating:** 2
**Confidence:** 4

**Summary:**

This paper addresses the stability gap problem in continual learning. The authors propose a biologically-inspired approach based on noradrenergic gain modulation, where neuronal gains are dynamically adjusted based on prediction uncertainty. The proposed NGM-SGD implements uncertainty-driven gain boosts that create fast-slow weight dynamics and flatten the loss landscape. The approach is evaluated on domain-incremental and class-incremental benchmarks under ideal joint training conditions, demonstrating reduced stability gaps compared to standard optimizers.

**Strengths:**

**Solid theoretical foundation**
The mathematical framework connecting gain modulation to fast-slow weight decomposition is clear and intuitive. The analysis showing how gain boosts flatten the loss landscape provides mechanistic insight.

**Clear, implementable algorithm.**
The application of NGM-SGD seems simple, with standard SGD weight update plus a gain update driven by prediction entropy each iteration. The lack of architectural changes, replay buffers, or extra losses makes it practical, and the idea that such a minimal mechanism can shrink the stability gap in continual learning is compelling.

**Weaknesses:**

**Novelty with respect to biological grounding.**
The authors claim that no prior work has adopted a bio-inspired approach to mitigate the stability gap or connected it back to adaptive biological learning. However, the complementary learning systems (CLS) literature has long modeled fast/slow learning via mimicking the hippocampus–neocortex interactions of the brain, and many continual learning methods explicitly borrow this paradigm through dual-memory architectures, replay-based consolidation, etc. [1, 2] The proposed gain-modulated fast/slow decomposition closely echoes this CLS framing. The manuscript should acknowledge these approaches, explain the similarities or distinctions, and include comparisons or at least a reasoned discussion against strong CLS-style baselines. Without this positioning, the biological novelty claim feels somewhat overstated.

**Limited empirical scope.**
While the paper notes the simplicity of its benchmarks as a limitation, this is not a minor caveat. It is a necessary extension to substantiate the paper. Without evaluations on more challenging settings (e.g., longer task streams, larger-scale datasets, online/streaming protocols without task IDs, and realistic memory/compute constraints) it is difficult to conclude that the method genuinely mitigates stability gaps rather than benefiting from the specifics of the setup. Also, comparisons should be made with recent continual learning methods specifically designed to address stability gaps. Expanding the study to stronger baselines and modern architectures would further strengthen the claim.

**Hyperparameter sensitivity**
The method introduces additional hyperparameters, requiring task-specific tuning. The values vary significantly across datasets, suggesting the method may not generalize well. A guidance on how to set these parameters for new tasks would be helpful.

References

[1] Arani, Elahe, Fahad Sarfraz, and Bahram Zonooz. "Learning fast, learning slow: A general continual learning method based on complementary learning system." arXiv preprint arXiv:2201.12604 (2022).

[2] Pham, Quang, Chenghao Liu, and Steven Hoi. "Dualnet: Continual learning, fast and slow." Advances in Neural Information Processing Systems 34 (2021): 16131-16144.

**Questions:**

1. The performance improvements seems modest and inconsistent. Why does vanilla SGD or other baselines sometimes outperform multi-timescale optimizers? This seems counterintuitive given your multi-timescale argument.

2. As mentioned, in CNN experiments, gain modulation is applied only to the output layer. What happens if you apply gain modulation to all layers in CNNs instead of just the output layer? Is there an adaptive way to slightly modify the methodology so that gain modulation can be applied to all layers of CNN?

---

> ### Author Response · Authors · 2025-11-17
>
> We thank the reviewer for the feedback. Please find the responses to your questions and concerns below (Weakness W, Question Q).
>
> **W1:** We apologise for any misunderstanding about the biological grounding of our approach. CLS methods indeed model fast-slow learning, but they do so through architectural modularisation and replay-based consolidation aimed at catastrophic forgetting. Our contribution operates at a different level: NGM-SGD modulates learning directly through a dynamic gain signal at the weight-update level, making it closer to a two-timescale optimiser than to a CLS-style system. Moreover, the cited CLS work predates the identification of the stability gap (May 2022, [1]) and evaluates only final performance, not the transient drops revealed under continual evaluation. Since these methods rely on replay and external modules, they are not directly comparable to our full-replay setting, which already represents the upper bound for replay-based approaches.
>
> **W2:** We agree that broader empirical validation is valuable, and we will clarify our experimental rationale more explicitly. We use joint training and short task streams deliberately to isolate the stability gap from other sources of forgetting. The setting is online and task agnostic (data streamed without task IDs), and the gain responds only to predictive entropy, naturally signalling task switches. While full replay is idealised, it provides the cleanest environment to study the optimiser’s contribution without interference from architectural or replay limitations.
>
> Regarding baselines, the stability gap was identified only recently [1] and early attempts to mitigate it were unsuccessful, pointing to its origin in the optimisation procedure itself [2]. Since NGM-SGD acts purely at the optimiser and weight-update level, the most meaningful baselines are standard optimisers, which underpin all CL methods. Other approaches rely on assumptions incompatible with our setting: classifier expansion across tasks [3], pretrained backbones [4], [5], or maintaining moving averages of task-specific weights [6] -- i.e. introducing architectural changes or task information. For these reasons, optimiser-level baselines are the appropriate comparison for the phenomenon we study, and NGM-SGD should be viewed as complementary to CL methods rather than competing with them.
>
> **W3:** Our method introduces three hyperparameters (learning rate, $\eta$, $\gamma$). In practice, $\gamma=0.9$, and only $\eta$ requires mild tuning. $\eta$ needs to keep the gain in a reasonable range, which depends on the dataset’s average output entropy: low-entropy datasets allow larger $\eta$, while high-entropy datasets require smaller values. For reference, we report the corresponding mean entropies ($\bar{H}$) in the table below.
>
> |               | $\bar{H}$                  | $\eta$ |
> |----------------------|-------------------------------------------|----------------|
> | Rotated MNIST        | $0.36$                      | $0.5$          |
> | Split MNIST          | $0.24$                      | $0.4$          |
> | Split CIFAR-10       | $0.54$                      | $0.2$          |
> | Domain CIFAR-100     | $1.06$                      | $0.1$          |
> | Split mini-ImageNet  | $1.65$                      | $0.1$          |
>
> **Q1:** We apologize for the original computation of avg-min-ACC and WC-ACC mistakenly included the first task, not reflecting plasticity-stability trade-offs (see the general comment). Vanilla SGD does not outperform our method, but it can exceed the other multi-timescale optimisers because it has no inertia term. As discussed in the results, MSGD and Adam updates may overshoot at task switches, reducing stability, whereas SGD remains better aligned with the true gradient.
>
> **Q2:** Applying gain modulation to all CNN layers disrupts feature learning and interferes with batch-normalisation, making training unstable. We therefore restrict gain modulation to the classification head, where it influences task transitions without affecting feature extraction and is sufficient to attenuate stability-gap effects [3].
>
> **References:**
>
> [1]: De Lange, M., van de Ven, G., and Tuytelaars, T. (2022). Continual evaluation for lifelong learning: Identifying the stability gap.
>
> [2]: Hess, T., Tuytelaars, T., and van de Ven, G. M. (2023). Two Complementary Perspectives to Continual Learning: Ask Not Only What to Optimize, But Also How.
>
> [3]: Łapacz, W., Marczak, D., Szatkowski, F., and Trzciński, T. (2024). Exploring the Stability Gap in Continual Learning: The Role of the Classification Head.
>
> [4]: Harun, M. Y., and Kanan, C. (2023). Overcoming the Stability Gap in Continual Learning.
>
> [5]: Guo, Y., Fu, J., Zhang, H., Zhao, D., and Shen, Y. (2024). Efficient Continual Pretraining by Mitigating the Stability Gap.
>
> [6]: Soutif-Cormerais, A., Carta, A., and van de Weijer, J. (2023). Improving Online Continual Learning Performance and Stability with Temporal Ensembles.

---

> > ### Comment · Reviewer_4gom · 2025-11-27
> >
> > The authors’ comments have resolved part of my concerns. However, I still believe that additional experiments are necessary. While I understand that NGM-SGD addresses the stability gap observed even under joint training, I think the method would become more practically convincing if it were also validated in more realistic CL settings. Are there experiments in a non-joint CL scenario?

---

> > > ### Author Response · Authors · 2025-12-02
> > >
> > > We thank the reviewer for noting that our previous response addressed part of their concerns, and we especially appreciate the acknowledgement that NGM-SGD effectively addresses the stability gap observed even under joint training. This recognition aligns with the central goal of our work.
> > >
> > > Regarding the request for more realistic CL settings, we emphasise that the experimental design in the main paper is intentional. In the revised manuscript, we have clarified this rationale in the Introduction and explicitly reiterated it in the *Limitations and future work* paragraph under the Discussion section. Our choice reflects two complementary considerations. First, using a small number of tasks ensures that each switch produces a substantial distributional shift, making the stability gap large enough to measure reliably. Second, the joint-training setup isolates the transient, optimisation-driven forgetting that defines the stability gap, avoiding confounds from steady-state forgetting caused by inefficient replay dynamics or inadequate loss regularisation. This is analogous to isolating a physical phenomenon under controlled laboratory conditions before adding additional forces or noise.
> > >
> > > However, we fully agree that examining NGM-SGD in more realistic continual learning conditions is valuable. To address this, we have **added a comprehensive set of experience replay (ER) experiments** in Appendix H. These experiments use class-balanced buffers and homogeneously mixed batches. In the case of small buffers, which is common in practical continual learning, the long-term loss of performance on earlier tasks can become larger than the transient drop that defines the stability gap (see Figure 9 in the revised manuscript). When this steady state forgetting exceeds the transient effect, it masks the stability gap and makes it difficult to measure reliably. This reinforces the motivation for our controlled setting, where the stability gap can be examined without interference from replay under-sampling or poor approximations of the joint loss. To reveal the gap more clearly, we also ran experiments with larger buffers that allow models to recover performance on earlier tasks, and under these conditions, the stability gap becomes measurable again. Across most benchmarks, NGM-SGD (our method) continues to reduce the gap more effectively than the baseline optimizers, showing that its gain dynamics remain beneficial even in more realistic continual learning scenarios (Table 6 in the revised manuscript).

---

### Official Review · Reviewer_CCxn · 2025-11-02

**Soundness:** 2
**Presentation:** 3
**Contribution:** 2
**Rating:** 2
**Confidence:** 2

**Summary:**

This paper addresses the stability gap in continual learning by introducing uncertainty-modulated gain dynamics as a two-timescale gradient descent adjustment  that balances adaptation and retention by adjusting learning rates and flattening the energy landscape. The authors demonstrate analytically that this neuron modulation gain induces fast-slow weight scales and flattens the local loss surface near the minima. The authors evaluate their method on MNIST, CIFAR, and mini ImageNet continual learning benchmarks against baseline optimizers: momentum-SGD, Adam, and SGD.

**Strengths:**

- Empirical evidences shows that NGM-SGD reduces test loss at task boundaries.
- Empirical evidence shows that NGM-SGD reduces the stability gap.

**Weaknesses:**

- See the first bullet point in the questions, why is it necessary to compare NGM-SGD only to other optimizers: SGD, Adam, MSGD? Could there not exist some continual learning method that outperforms MSGD in the metrics illustrated in Table 1? Given this lack of a comparison to existing continual learning methods, why do the results support the efficacy of NGM-SGD?
- Overall, the empirical results are mixed, see Table 1. For instance, the baseline optimizers attain comparable if not often better performance on many of the reported metrics, than NGM-SGD. While some clear benefits of NGM-SGD are observed, the overall mixed results and limited scale and scope of the experiments puts into question the efficacy of the method.
- While the biological motivation well motivates the proposed method, for many readers parsing this information can be difficult. It would be useful for many readers if the algorithm and contributions were distilled algorithmically, rather than solely being motivated from a neurological phenomenon, earlier in the paper.

**Questions:**

- Why are only SGD, Adam, and MSGD evaluated against NGM-SGD? Could there be existing CL methods that outperform NGM-SGD and would be worthwhile comparing? How should we think of NGM-SGD interfacing with other continual learning interventions?

---

> ### Author Response · Authors · 2025-11-17
>
> We thank the reviewer for the insightful feedback. Please find the responses to your questions and concerns below (Weakness W, Question Q).
>
> **W1 & Q1:** We thank the reviewer for raising this point. The stability gap has been identified only recently [1], and the first attempt to tackle it was unsuccessful [2], suggesting that the stability gap’s origin lies in the optimisation procedure itself. Our focus is therefore on understanding whether the optimisation dynamics contribute to the stability gap. Since NGM-SGD operates purely at the optimiser and weight-update level, the most meaningful baselines are the standard optimisers (vanilla SGD, MSGD, Adam) that underlie all continual learning methods.
>
> We acknowledge that other works have been proposed to target this problem, but they rely on assumptions incompatible with our setting: [3] expand the classifier at every task (increasing complexity and not being task agnostic), [4] and [5] rely on pretrained backbones and LoRA-style updates (also not task agnostic), and [6] maintains a smoothed average of the weights during task transitions (computationally heavy and not task agnostic). These approaches introduce architectural changes or external modules, whereas our method operates purely at the optimiser and weight-update level.
>
> For this reason, optimiser-level baselines are the appropriate comparison for the phenomenon we study in our online task-agnostic setting. Our contribution concerns how the model is optimised, not what additional CL machinery is added, and we view NGM-SGD as complementary to existing CL interventions rather than competing with them.
>
> **W2:** We thank the reviewer for pointing this out. The original computation of avg-min-ACC and WC-ACC mistakenly included the first task, i.e., the model’s initialization before any task transition. Since the stability gap manifests only after transitions, this made the second and third columns of Table 1 misleading. We have recomputed these metrics using only task $k \geq 2$, which correctly isolates the plasticity–stability effects. The updated table (to be included in the revised manuscript, see the general comment) shows a clearer improvement of NGM-SGD on these metrics, consistent with the behaviour observed in the figures. Furthermore, Table 4 (in the appendix) reports the stability gap at each task transition and shows a clear attenuation of these transient drops under NGM-SGD. Furthermore, our method mitigates the stability gap (the primary aim of this work) while maintaining similar overall performance.
>
> **W3:** We apologise for this. In the current version, we leaned strongly on the biological motivation to provide context, which may have made the exposition harder to follow. We will revise the presentation and reframe the discussion to place greater emphasis on the ML relevance of the method, introducing the algorithmic contributions earlier and more directly. The core mechanism is that the gain induces a reparametrization of the weights, which flattens the effective loss landscape during evaluation (in line with prior work on how reparametrizations alter curvature, see [7]), together with an adapted step size in the backward pass. This makes NGM-SGD behave differently from methods that only adjust the learning rate, since those only affect the backward update and do not modify the geometry of the forward landscape. We will make this clearer in the revised manuscript.
>
> **References**
> [1]: De Lange, M., van de Ven, G., and Tuytelaars, T. (2022). Continual evaluation for lifelong learning: Identifying the stability gap. arXiv:2205.13452.
>
> [2]: Hess, T., Tuytelaars, T., and van de Ven, G. M. (2023). Two Complementary Perspectives to Continual Learning: Ask Not Only What to Optimize, But Also How. arXiv:2311.04898.
>
> [3]: Łapacz, W., Marczak, D., Szatkowski, F., and Trzciński, T. (2024). Exploring the Stability Gap in Continual Learning: The Role of the Classification Head. arXiv:2411.04723.
>
> [4]: Harun, M. Y., and Kanan, C. (2023). Overcoming the Stability Gap in Continual Learning. arXiv:2306.01904.
>
> [5]: Guo, Y., Fu, J., Zhang, H., Zhao, D., and Shen, Y. (2024). Efficient Continual Pretraining by Mitigating the Stability Gap. arXiv:2406.14833.
>
> [6]: Soutif-Cormerais, A., Carta, A., and van de Weijer, J. (2023). Improving Online Continual Learning Performance and Stability with Temporal Ensembles. arXiv:2306.16817.
>
> [7]: Dinh, L., Pascanu, R., Bengio, S., & Bengio, Y. (2017). Sharp Minima Can Generalize For Deep Nets. ICML

---

### Official Review · Reviewer_NrGM · 2025-11-06

**Soundness:** 2
**Presentation:** 2
**Contribution:** 3
**Rating:** 4
**Confidence:** 3

**Summary:**

The work introduces a method that effectively decomposes weights into two components, a slow component and a fast component. This is implemented as a neuronal gain that is multiplied to the weights of the neural network. At each step, this gain is decayed to some base value and increased based on the neural network’s uncertainty, which is biologically inspired by noradrenaline. The paper shows that their approach mitigates the “stability gap” that occurs in continual learning when switching tasks.

**Strengths:**

- The idea of using gain modulation as a flexible way to handle distribution shifts, and showing it can help with the stability gap is good, and it is empirically validated in the supervised continual learning experiments that are presented.
- Neuronal gain as a proxy for task complexity is interesting. The results make sense as the neuronal gain is essentially moving average of the entropy of the outputs.

**Weaknesses:**

- Overall, while the method does result in an optimizer that mitigates the stability gap, it does seem to do that at the expense of overall performance.
- The proposed method has significantly more hyperparameter configurations evaluated compared to the baselines (15x more). This could very easily be the reason for any performance gains of NGM-SGD.
- I am not sure leaning so heavily into the biological framing is useful/correct. One of the contributions is listed as “We link our algorithmic gain bursts to noradrenergic neuromodulation” but there is not really much in the paper linking what happens in the biology to what’s happening in the artificial networks, other than a vague notion of uncertainty. Saying it’s biologically inspired is fine, but framing one of the contributions of your paper as “noradrenergic neuromodulation” seems like overclaiming. It’s also unclear if what is used for the uncertainty proxy in the paper (the entropy of the softmax) is valid given the networks are uncalibrated.
- I think a bit more work should be done on the dynamics of how the gain evolves with how the output evolves. Specifically, if the gain goes up, the norm of all weights increases, how does that affect the effective learning rate?

**Questions:**

- The flattening effect that you mention, isn’t it mitigated by the network taking effectively smaller steps? Could you give more detail into how the trajectory followed by the optimizer would look like the trajectory in a flattened minima?

---

> ### Author Response · Authors · 2025-11-17
>
> We thank the reviewer for the insightful feedback. Please find the responses to your questions and concerns below (Weakness W, Question Q).
>
> **W1:** We thank the reviewer for pointing this out. The original computation of avg-min-ACC and WC-ACC mistakenly included the first task, i.e., the model’s initialization before any task transition. Since the stability gap only arises after transitions, this inclusion made the second and third columns of Table 1 misleading. We have recomputed these metrics using only tasks $k \geq 2$, which correctly isolates plasticity-stability effects. The updated table (which will be included in the revised manuscript, see general comment) shows the expected improvement of NGM-SGD on these metrics, consistent with the figures.
>
> Regarding avg-ACC, NGM-SGD remains within the error bars of the other optimizers (with a small exception on Rotated MNIST vs. Adam). This shows that our method mitigates the stability gap while maintaining similar overall performance. As discussed in the manuscript, reducing the stability gap does not necessarily translate into higher final accuracy, despite this possibility being suggested in prior work [1].
>
> **W2:** We would like to clarify that NGM-SGD does not rely on a larger or higher-dimensional hyperparameter search space than the baselines (Adam, vanilla SGD, MSGD). Our method uses three hyperparameters (learning rate, $\eta$, $\gamma$), which is comparable to Adam [2] (learning rate, $\beta_1$, $\beta_2$, $\epsilon$) and only one more than MSGD [3] (learning rate, momentum). In our work, $\gamma$ is very robust across datasets (hyperparameter searches consistently yield 0.9), and only $\eta$ remains a free parameter ($\eta \in$ [0, 0.5]), as it reflects the magnitude of the distribution shift in each benchmark. Therefore, the observed stability gap performance gain of NGM-SGD is not a consequence of disproportionate hyperparameter tuning.
>
> **W3:** We thank the reviewer for the comment. We believe the biological framing is both useful and appropriate, as noradrenergic systems are well known to regulate neuronal gain under uncertainty (in our case entropy), providing a principled motivation for why gain modulation can help the integration of new information (e.g., [4]; [5]). Also, the use of entropy as an uncertainty proxy follows classical work on unexpected uncertainty in neuroscience [6] and experimental work has already determined that uncertainty proxy boosts gain [4] (see Eq 2. of our submission).
>
> We are unclear as to the use of the word “uncalibrated” in this context. If this refers to a network that has not encountered a specific stimulus before, this uncalibration is important to identify distribution shifts and task transitions.
>
> **W4 & Q1:** Thank you for these observations. When the gain increases, it does scale the effective learning rate during backpropagation, but its effect is not equivalent to taking smaller or more conservative steps. The key mechanism is the reparametrization of the forward weights, $W_\text{eff}=g w$, which modifies the curvature of the loss surface in the effective parameter space without changing the represented function. As shown in prior analyses of parameter reparametrization (e.g., [7]), such transformations can locally flatten the landscape. Under NGM-SGD, the optimizer therefore moves through a region that behaves as if it were flatter in the $W_\text{eff}$ space, even though the step size in $w$-space may increase when gain rises. This dual effect of amplifying gradient steps while simultaneously reducing curvature (Hessian) in the effective space is what stabilizes task transitions and cannot be achieved by learning-rate adjustments alone. We will add a theoretical clarification of this mechanism in the Appendix.
>
> **References**
>
> [1]: Hess, T., Tuytelaars, T., and van de Ven, G. M. (2023). Two Complementary Perspectives to Continual Learning: Ask Not Only What to Optimize, But Also How. arXiv:2311.04898.
>
> [2]: Kingma, D. P., and Ba, J. (2014). Adam: A Method for Stochastic Optimization. arXiv:1412.6980.
>
> [3]: Qian, N. (1999). On the momentum term in gradient descent learning algorithms. Neural Networks, 12(1), 145–151.
>
> [4]: Wainstein, G. et al. (2025). Gain neuromodulation mediates task-relevant perceptual switches. eLife.
>
> [5]: Munn, B. R. et al. (2023). Neuronal connected burst cascades bridge adaptive signatures across arousal states. Nat. Commun., 14, 6846.
>
> [6]: Angela J. Yu and Peter Dayan. Uncertainty, Neuromodulation, and Attention. Neuron, 46(4): 681–692, May 2005. ISSN 08966273. doi: 10.1016/j.neuron.2005.04.026
>
> [7]: Dinh, L., Pascanu, R., Bengio, S., & Bengio, Y. (2017). Sharp Minima Can Generalize For Deep Nets. ICML.

---

### Author Response · Authors · 2025-11-17
**General Comment**

We thank all reviewers for their detailed feedback and apologise if parts of the manuscript did not clearly convey the main aims of our work. Several concerns appear to stem from misunderstandings about the scope and focus of the paper. We have distilled the paper to the following main points:

- **Problem statement.** Recent continual learning (CL) studies reveal a stability gap -- a temporary drop in performance on past tasks when new ones are introduced -- highlighting an imbalance between rapid adaptation and long-term retention. This work introduces an adaptive gain mechanism that acts as a multi-timescale, task-agnostic optimizer by splitting weight updates into fast and slow components. This mechanism also induces a weight reparametrization that alters the effective curvature of the loss landscape during the forward pass, leading to smoother and more stable gradient steps during learning and effectively reducing the stability gap in online continual learning settings. We specifically test this under full-replay continual learning benchmarks to isolate the stability gap from other sources of forgetting, and observe that our method consistently attenuates the stability gap across benchmarks.

- **Scope of benchmarks.** We use joint training and small task streams intentionally to isolate the stability gap from other sources of forgetting and study its origin in a clean setting. Our goal is to analyse how the optimiser contributes to transient instabilities, not to propose a full CL system. We will make this distinction explicit.

- **Choice of baselines.** Our method operates at the optimiser and weight-update level, so standard optimisers (SGD, MSGD, Adam) are the most appropriate baselines. Recent stability-gap methods rely on architectural changes, pretrained models, or task-dependent components that are not directly comparable to our online task-agnostic setting.

We appreciate the reviewers' insights and will revise the manuscript to clarify the aims of the study, improve the ML-oriented framing, and better articulate how NGM-SGD fits within the broader continual-learning landscape. We believe these revisions will significantly improve the clarity and impact of the work.

Additionally, we would like to highlight updated performance gains in Table 1. We acknowledge that the original computation of avg-min-ACC and WC-ACC mistakenly included Task 1, which does not reflect plasticity-stability trade-offs. Recomputing these metrics for tasks $k=2, ..., T$, shows better performance of NGM-SGD against other optimizers, consistent with the reported figures. avg-ACC and avg-SG remain the same as initially presented. The corrected Table 1 will be included in the revised manuscript.

| Benchmark            | Method   | avg-min-ACC           | WC-ACC               |
|----------------------|----------|------------------------|-----------------------|
| Split MNIST          | NGM-SGD (ours)  | **97.570 ± 0.465**     | **97.696 ± 0.373**    |
|                      | MSGD     | 72.518 ± 0.403         | 77.628 ± 0.329        |
|                      | Adam     | 71.047 ± 3.753         | 76.537 ± 3.003        |
|                      | SGD      | 95.887 ± 2.887     | 96.293 ± 2.311    |
| Split CIFAR-10       | NGM-SGD (ours)  | **79.485 ± 3.875**     | **80.880 ± 3.349**    |
|                      | MSGD     | 52.828 ± 3.898         | 58.348 ± 3.521        |
|                      | Adam     | 63.450 ± 7.839         | 67.692 ± 6.335        |
|                      | SGD      | 68.712 ± 4.958         | 71.868 ± 4.007        |
| Split mini-ImageNet  | NGM-SGD (ours)  | **33.165 ± 2.435**     | **36.040 ± 2.058**    |
|                      | MSGD     | 26.775 ± 2.417         | 30.416 ± 2.029        |
|                      | Adam     | 28.300 ± 3.371     | 32.052 ± 2.768    |
|                      | SGD      | 28.345 ± 3.671     | 30.288 ± 3.309        |
| Rotated MNIST        | NGM-SGD (ours)  | **90.015 ± 1.451**     | **91.542 ± 0.973**    |
|                      | MSGD     | 84.329 ± 1.359         | 87.823 ± 0.914        |
|                      | Adam     | 87.177 ± 1.060         | 89.910 ± 0.721    |
|                      | SGD      | 89.107 ± 1.935     | 90.787 ± 1.300    |
| Domain CIFAR-100     | NGM-SGD (ours)  | **53.420 ± 2.804**     | **58.290 ± 1.916**    |
|                      | MSGD     | 51.370 ± 2.234     | 56.867 ± 1.556    |
|                      | Adam     | 51.615 ± 2.653     | 57.323 ± 1.840    |
|                      | SGD      | 49.050 ± 3.177     | 54.950 ± 2.229    |

---

### Meta-Review · Area_Chair_ifub · 2026-01-07

**Summary:**

None of the reviewers suggested accepting this submission. It received the ratings of 4,2,2,4,4. Authors provided response and 3 of the reviewers (ratings: 2,4,4) interacted with the authors. None of those 3 reviewers were convinced enough by the authors' response to improve their rating.

**Reviewer Concerns:**

The main concerns are: lack of a comparison to existing continual learning methods, the empirical results are mixed, significantly more hyperparameter configurations compared to baselines, over-claiming on biological inspiration, and novelty.

**Reviewer Scores:**

4,2,2,4,4.

3 of the reviewers already interacted with the authors and were not convinced enough to improve their ratings. Also reading the reviews and author responses, the concerns are important enough that I do not think the author responses would have resolved them.

---

### Decision · Program_Chairs · 2026-01-26

Reject